# Genome analysis in *Avena sativa* reveals hidden breeding barriers and opportunities for oat improvement

Nicholas A. Tinker [1✉], Charlene P. Wight [1], Wubishet A. Bekele[1], Weikai Yan[1], Eric N. Jellen [2], Nikos Tsardakas Renhuldt [3], Nick Sirijovski [3,4,8], Thomas Lux [5], Manuel Spannagl [5] & Martin Mascher [6,7]

Oat (*Avena sativa* L.) is an important and nutritious cereal crop, and there is a growing need to identify genes that contribute to improved oat varieties. Here we utilize a newly sequenced and annotated oat reference genome to locate and characterize quantitative trait loci (QTLs) affecting agronomic and grain-quality traits in five oat populations. We find strong and significant associations between the positions of candidate genes and QTL that affect heading date, as well as those that influence the concentrations of oil and β-glucan in the grain. We examine genome-wide recombination profiles to confirm the presence of a large, unbalanced translocation from chromosome 1 C to 1 A, and a possible inversion on chromosome 7D. Such chromosome rearrangements appear to be common in oat, where they cause pseudo-linkage and recombination suppression, affecting the segregation, localization, and deployment of QTLs in breeding programs.

[1] Agriculture and Agri-Food Canada, Ottawa Research and Development Centre, 960 Carling Avenue, K.W. Neatby Bldg., Central Experimental Farm, Ottawa K1A 0C6 ON, Canada. [2] Department of Plant and Wildlife Sciences, Brigham Young University, 4105 LSB, Provo 84602 Utah, USA. [3] Lund University, Department of Chemistry, Division of Pure and Applied Biochemistry, Box 124, 221 00 Lund, Sweden. [4] CropTailor AB, c/o Lund University, Department of Chemistry, Division of Pure and Applied Biochemistry, Box 124, 221 00 Lund, Sweden. [5] Helmholtz Center Munich – Research Center for Environmental Health, Plant Genome and Systems Biology (PGSB), Ingolstaedter Landstr. 1, 85764 Neuherberg, Germany. [6] Leibniz Institute of Plant Genetics and Crop Plant Research (IPK), Domestication Genomics, Corrensstrasse 3, 06466 Seeland, Germany. [7] German Centre for Integrative Biodiversity Research (iDiv) Halle-Jena-Leipzig, Puschstrasse 4, Leipzig, Germany. [8]Present address: Oatly AB, Food Science, Scheelevägen 19, 223 63, Lund, Sweden. ✉email: Nick.Tinker@agr.gc.ca

Cultivated hexaploid oat (*Avena sativa* L.) is a cereal crop of international importance, widely considered to be a healthy and nutritious whole grain food. The improvement of cultivated oat varieties used for food focuses primarily on agronomic traits, with some emphasis on increasing the concentration of (1, 3; 1, 4)-ß-D-glucans (hereafter, β-glucan) and on lowering the oil concentration in the grain. These two seed-related traits allow oat processors to maintain a nutritional profile that meets labeling requirements for various health claims[1]. The oat milling industry also values traits related to grain handling (e.g., test weight) and milling yield (the proportion of groats recovered after de-hulling). However, with the growing interest in new oat-based products such as dairy and meat substitutes, there may be an increased need to develop oat varieties that meet unique seed quality profiles such as increased protein and oil content.

Previous QTL studies have not benefited from a complete annotated reference genome, and those published before 2016 suffered from a lack of high-density marker loci. The use of different marker systems and the presence of physical rearrangements among oat chromosomes have also made it challenging to compare QTL results among different experiments. With the recent availability of a high-density SNP consensus map[2,3], sequenced diploid *Avena* relatives[4], and a fully-annotated hexaploid reference genome of the spring oat cultivar 'Sang' (Kamal et al., in press)[5], updated assessments of QTLs affecting agronomic traits and grain quality in the context of comparative genomics can now be made.

Previously, high variation in kernel quality and nutritional traits was described in two recombinant inbred line (RIL) populations from a common parent 'HiFi' crossed with 'Goslin' and 'Sherwood' (crosses abbreviated hereafter as GoHf and ShHf)[6]. HiFi has high β-glucan and oil, but its seed has high hull content and is hard to de-hull, while Sherwood and Goslin are extreme in the opposite direction. Goslin is a parent of Sherwood, and both varieties have excellent grain-milling characteristics but are daylength-insensitive, making them less adapted to northern regions where long days trigger daylength-sensitive oat varieties to flower early. Previously, both the GoHf and ShHf populations lacked molecular marker data to locate QTLs affecting these traits.

Two other populations that have high-quality phenotypic data for oat quality and agronomic performance include 'Dal' x 'Exeter'[7] and 'Terra' x 'Marion'[8]: abbreviated hereafter as DaEx and TeMa. Dal has high oil content, while Terra is a hulless line. Exeter and Marion are older varieties that were grown widely for their adaptation to northern production environments. While previous QTL studies were performed in both DaEx and TeMa, these studies lacked high-density sequence-based markers that could be positioned relative to a reference genome. A further cross, 'TX07CS-1948' (a southern-adapted line from Texas, USA) x 'Hidalgo' (a spring variety from Saskatoon, Canada) was studied previously to identify QTLs for height, heading date, and disease resistance[9]. This population, designated hereafter as TxHd, showed high quantitative trait variation, especially for heading date. However, it was noted that further work should be performed to identify the reason for lack of recombination on linkage group 'Mrg02' (now chr7D) on which major QTLs affecting heading data and rust resistance were located[9]. We note here that the previous 'Mrg' designations refer to linkage groups[2,3] while our use of the abbreviation 'chr' followed by a number (1 through 7) and a sub-genome designation (A, C, or D) indicates a hexaploid oat chromosome nomenclature that has been approved by the International Oat Nomenclature Committee (https://wheat.pw.usda.gov/GG3/oatnomenclature).

The primary objectives of the current work were: (1) to develop new high-density sequence-based marker data in five oat RIL populations; (2) to develop an expanded population size with additional phenotype data in TxHd; (3) to develop reference-guided QTL and candidate gene analyses of traits affecting seed quality and agronomic performance, and (4) to characterize recombination profiles relative to a reference genome in order to identify anomalies and potential chromosome rearrangements that have impacts on the introgression and deployment of QTLs in practical oat breeding programs. In summary, we found strong QTL affecting many traits in all five populations. We found strong and significant associations between the positions of candidate genes and QTL that affect heading date, as well as those that influence the concentrations of oil and β-glucan in the grain. The recombination profiles relative to the reference genome revealed a large, unbalanced translocation from chromosome 1C to 1A in two populations, and a possible inversion on chromosome 7D in TxHd.

## Results

**Marker analysis and imputation**. Full marker data sets for all five populations were developed for data where markers were sorted by position in the reference genome, but non-imputed (Supplementary Data 2), data imputed by FSFHap (Supplementary Data 3), and data imputed by GBSi (Supplementary Data 4). The total number of markers in the non-imputed data for populations DaEx, GoHf, ShHf, TxHd, and TeMa was 5579, 7535, 10,535, 7870, 2895, respectively. For the FSFHap and GBSi data, these numbers were slightly reduced by approximately the same amount, resulting in 5478, 7495, 10,475, 7824, and 2856 markers vs. 5485, 7505, 10,495, 7838, and 2864 markers (in the same order, respectively). The slight reductions were because of different markers being dropped by the imputation algorithms for lack-of-fit. However, the FSFHap imputation method was not able to impute data on the short chromosome region of the chr1C → chr1A translocation (see below) and on chr7D in TeMa, where only 32 markers were available. Other artefacts were apparent when this imputation method was used in regions where marker density was low (e.g., chr1D in TeMa). For this reason, and because both the GBSi and FSFHap-imputed data sets gave very similar QTL analyses (compared within Supplementary Data 5), we chose to use the GBSi data set (Supplementary Data 4) in further analyses. We provide the non-imputed and FSFHap-imputed data (Supplementary Data 3 and 4) only for the purposes of comparison.

Based on locations of GBS markers in Sang compared to matches of GBS markers in the A and C genomes, it was determined that a translocated region beginning near 420 Mbp on chr1A, and covering the remaining 106 Mbp of the distal part of chr1A, originated from ancestral chr1C. A very small reciprocal translocation on chr1C was observed from 422 Mbp and covering only the 5 Mbp distal part of chr1C. Dot-plots based on genome alignments between Sang, *A. atlantica*, and *A. eriantha* were used to refine the position of the chr1C → chr1A translocation breakpoint to between 420,017,662 and 420,180,594 bp on Sang chr1A (Supplementary Fig. 1). Similarly, the breakpoint of the small chr1C → chr1A reciprocal translocation segment was estimated to be between 421,703,092 bp and 421,709,035 on Sang chr1C (Supplementary Fig. 2). To highlight these chromosome regions, and to avoid potential artefacts in marker imputation, both regions were analyzed henceforth as if they were separate chromosomes (named chr1AC and chr1CA, respectively), but the bp coordinates from the original Sang chromosomes were preserved for ease of reference. This modification is also reflected in the GBS sort table (Supplementary Data 2).

**Chromosome recombination landscapes**. Average pairwise recombination rates among all chromosomal regions were

computed in 16 Mbp sliding windows at 10 Mbp increments for each population (Supplementary Figs. 3 to 7) and in 1 Mbp increments for chromosomes 1A and 1C of GoHf and ShHf (Supplementary Fig. 8). This method gave recombination matrices that were scaled by physical distance rather than being scaled by marker density. Although this provides a more accurate picture of physically-referenced recombination rates, several regions appear blocky because small numbers of markers were used to represent large physical distances (e.g., chr7D in GoHf and ShHf). Most chromosomes in most populations showed recombination suppression towards the centromere, with recombination hotspots near the telomeres. However, many anomalies in this pattern also were observed.

The most striking anomaly in the recombination matrices was a pattern of linkage between chr1A and chr1C in the ShHf and GoHf populations (Supplementary Fig. 8), which likely resulted from the previously known chr1C → chr1A translocation event. Analysis by C-banding (Supplementary Fig. 9) showed that HiFi does not contain this translocation, while Goslin does. Analyses of C-banding patterns were not yet available for other mapping parents, but the translocated chromosome configuration is known to be common in most spring oat varieties and many winter varieties[10]. Since GoHf and ShHf were the only populations that showed this recombination pattern, we conclude that HiFi is the only mapping parent in this study that does not contain this translocation. The pseudo-linkage pattern from ShHf (the larger of the two HiFi populations) is interpreted in Fig. 1a, b and

Supplementary Fig. 10. A simplified summary of this model is that all regions of chromosomes 1A and 1C tend to segregate together as parental haplotypes except for a long region on the non-translocated arm of chr1A, a short region on the non-translocated arm of chr1C, and the distal part of the chr1C → chr1A translocation, where recombination within a T- or unequal cross-shaped meiotic quadrivalent allows the segregation of gametes that produce either balanced and viable zygotes or gametes having very small duplications-deficiencies. In this model, the chr1C → chr1A translocation remains linked to the other regions of pseudo-linkage, either because it is too short to allow inter-chromosome pairing, or because recombinations are too rare to be observed. Although it is beyond the scope of this study to model or interpret the details of gamete selection or lethality, we note that large regions of both 1A and 1C, as well as the translocated regions, were distorted toward Sherwood alleles in ShHf, whereas only one small distorted region was observed in GoHf (Supplementary Data 4). This could signal a selection against HiFi gametes that occurs following the meiotic quadrivalent of ShHf.

A region of pronounced recombination suppression was also observed in the TxHd cross on chr7D. Specifically, no recombination was observed on this chromosome except on the distal (lower) telomeric region, despite an abundance of markers on this chromosome. Other populations also showed recombination suppression on this chromosome arm (Supplementary Figs. 3–6) but the TxHd suppression was most pronounced.

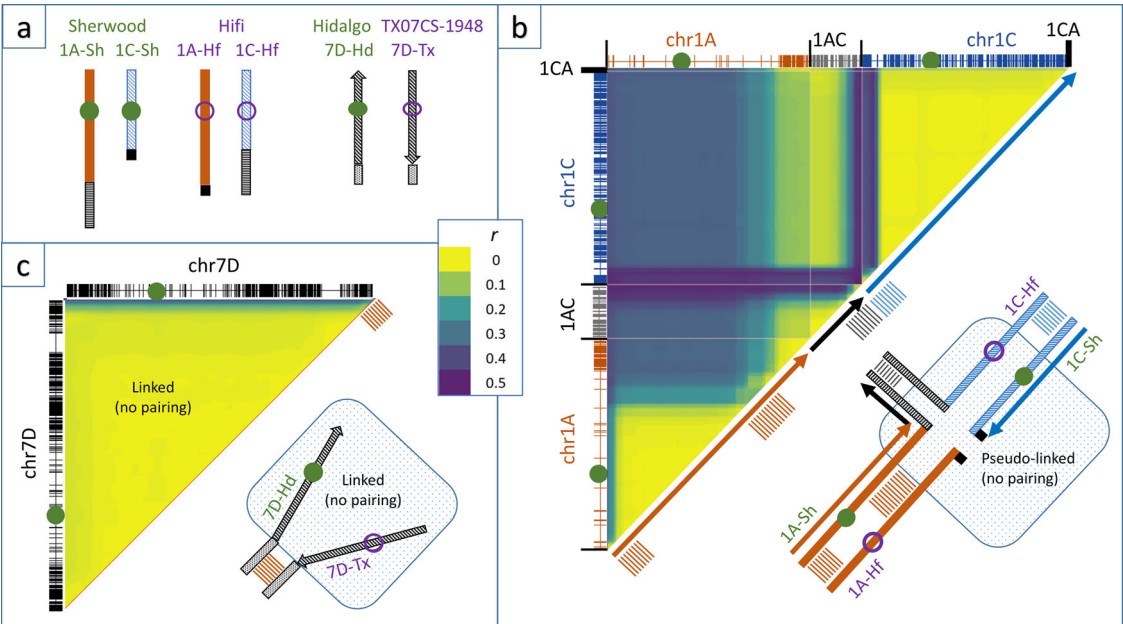

**Fig. 1 Interpretation of chromosome rearrangements leading to recombination suppression and pseudo-linkage.** Interpretations are based on empirical recombination data in the population 'Sherwood' crossed with 'HiFi' (ShHf; 215 progeny), and in 'TX07CS-1948' crossed with 'Hidalgo' (TxHd; 515 progeny). **a** schematic coding of relevant chromosomes 1A, 1C, and 7D in four parents, including the unbalanced chr1C → chr1A translocation (black parallel hatches and solid boxes) in Sh vs. Hf, and inverted (diagonal striped) vs. non-inverted (speckled) regions of chr7D in Tx vs. Hd. **b** An hypothetical quadrivalent meiotic pairing in an ShHf F₁ that would allow recombination between the non-translocated top arms of chr1A and chr1C as well as between the translocated and non-translocated chr1C → chr1A segment. For simplicity, only one-half of the chromatids are shown. The adjacent triangular heat map compares the observed recombination matrix between all chromosome regions computed in moving 16 Mbp windows. Marker positions are shown on the axes. The average recombination rates (r) between pairs of markers are visualized as blended colors of yellow (r = 0) to teal (r = 0.2) to burgundy (r = 0.5). Blocks of yellow indicate recombination suppression (within a chromosome) or pseudo-linkage (between chromosomes). Pairs of positions where recombination is observed are identified by burgundy. These positions are indicated by parallel lines on the diagonal of the heat map and on the corresponding regions of the meiotic quadrivalent. **c** Similar interpretation for a meiotic pairing of chr7D of a TxHd F₁. Here we observe recombination only on the bottom telomeric region, suggesting the presence of a large pericentric inversion of the entire remaining chromosome that prevents pairing and chiasma. It is possible that multiple inversions or other segmental rearrangements exist in this region, preventing the chromosomes from pairing in an opposite configuration.

A probable explanation for this suppression would be the presence of a large pericentric chromosome inversion between the parents of this population, as depicted in Fig. 1c. It is also possible that multiple inversions or other segmental rearrangements are present in this region, which would help to explain why the chromosomes would not simply pair in an opposite configuration. A heterogeneous series of segmental rearrangements could also explain why differing degrees of suppression are seen among different populations. Rearrangements in this region were not visible via C-banding (Supplementary Fig. 9) because diagnostic C-bands on chr7D are lacking. Currently, we have no preliminary data to interpret which parent of TxHd has the ancestral (non-inverted) configuration, and full characterization of this phenomenon will only be possible once additional reference genomes are available.

**QTL analysis**. Full QTL results for peaks exceeding LOD 3 (an approximate genome-wise error rate of 5%) for all populations, for each trait, site, and year are presented in Supplementary Data 5. Similarly, QTLs for trait means are in the same supplement, along with raw and interpreted matches of QTL candidate genes. These QTLs are provided in GFF format, so that they can be easily loaded in any genome browser for comparison to other genomic features, as demonstrated in a panel within Supplementary Data 5. While these formatted results will provide a rich source of information for future detailed comparative analysis, the results are too numerous to fully interpret here. Thus, we will focus our discussion on Table 1, which describes QTL regions based on analyses of trait means having LOD values that exceed 5. This LOD threshold was found by permutation analysis to provide a genome-wide error rate of ~0.001 (Supplementary Data 5). In Fig. 2, the positions of these major QTL are visualized in relation to candidate genes on a scaled version of the Sang reference genome.

Major QTLs (Table 1 and Fig. 2) were found for only a subset of the traits, and many QTLs were detected only in one or two of the five populations. In part, this is due to the limited number of populations where some traits were tested, but it is also not expected that every QTL will segregate in every biparental population, which was a reason for including five populations in this study. By relaxing the threshold and examining environment-specific QTL within the locations of major QTL, support from additional populations was provided for several of the QTL locations (see Frequency in Table 1). Overall, the traits for oil, β-glucan, and heading date had the strongest and most consistent QTL. The expansion of the TxHd population from 178 to 515 RILs provided a test of a large population to increase the power to detect QTLs. When compared to the previous study[9], this increase in population size detected all of the same QTL regions except for the height QTL on chr3C (Mrg15) that was previously found in only one environment (Supplementary Data 5). Of the remaining QTL, the current study detected these in an increased number of environments with LOD values that were, on average, 165% larger than in the smaller populations. Thus, although no new QTL regions were found in TxHd, we have validated all four major QTL regions for heading date, and three for height, using a population size that is much larger than that used in most QTL studies. It is noteworthy that this population has extreme transgressive segregation for both traits[9], and that positive QTL for each trait are found in both parents.

**Candidate genes and historical QTLs**. Candidate gene analysis was performed for CslF genes 3 through 12 (for β-glucan), ACC (for oil), and the heading-related genes VRN1, VRN3, and CO1. Supplementary Data 6 includes raw BLAST results for the Sang genome and its gene models and an interpretive summary of candidate gene positions. Supplementary Data 7 contains lists of QTLs described in previous studies that co-locate with the major QTLs listed in Table 1 and Supplementary Data 5, as well as with the candidate genes listed in Supplementary Data 6.

Of the three traits having putative candidate genes, there was a high degree of overlap between the sets of major QTL positions and the sets of tested candidate genes. In each case, the probability of detecting an overlap equal to or greater than that which was observed was smaller than 0.005 (Table 2). It is interesting that the bootstrap probabilities for β-glucan QTL were lower than those predicted by the binomial, while those for oil and heading date were higher. This suggests that gene densities are higher in the QTL regions for oil and heading date, and lower in the QTL regions for β-glucan. While this demonstrates a very high probability of association between the detected QTLs and tested candidate genes, it does not demonstrate a causal relationship, since other genes (including transcription factors) may follow the same sets of associations.

Overall, the traits that showed the most co-location between current and historical studies all relate to height, heading, and β-glucan, or were associated with the N1 gene in the TeMa population. It should be noted that the search for historical QTL was not intended to be exhaustive, but rather to investigate whether there was prior evidence for QTL at the locations found in the current study, and to alert the reader to literature related to particular QTLs. Historical studies take the form of biparental populations (where not all QTL are expected to be present) as well as GWAS studies. Neither have been conducted at a consistent level of significance. Thus, the quantification of the search space was not feasible for historical QTL, as it was for the candidate genes, and the statistical probability of historical QTL co-location has not been calculated. While many of the QTLs found in the current work do not co-locate with historical QTLs, this may be attributed to the small number of populations where some traits are measured, including in the current study, as well as to a lower heritability or a higher number of loci that affect certain traits.

All three of the major QTL for β-glucan were associated with candidate genes, with a probability of 0.0018 based on a simulation of randomly placed candidate genes (Table 2). Of the 11 β-glucan candidate gene locations that were tested, three were not associated with any QTLs, current or historical, while those associated with QTL in the current study (CslF11_chr6A_298Mb, CslF9_chr6A_416Mb, and CslF6_chr7A_399Mb) were associated with numerous historical QTLs in the populations 'Kanota' x 'Ogle' and Kanota x 'Marion' as well as in the CORE and UFRGS diversity panels (Supplementary Data 7). In the current study, all three of these regions contained positive (β-glucan-increasing) alleles from HiFi, a variety widely recognized as a founding cultivar for increased β-glucan. The two major HiFi QTL on Chr6A coincide with CslF9 and 11. However, evidence from barley suggests that these two genes are not involved in the expression of seed β-glucan[11,12], thus caution should be used in the interpretation of these genes as candidates, and other regulatory factors may be involved.

Of the three ACC loci, those on chromosomes 4C and 6A were associated with major oil-increasing alleles from HiFi and/or Dal, while that on 5D was not associated with oil QTL in the current study. There is some uncertainty about the chromosome identity of previously-detected QTL associated with ACC. One major oil QTL associated with an ACC clone was consistently found in an area of reduced recombination on linkage group (LG) 11 of the Kanota x Ogle and Kanota x Marion maps[13], and a second locus was found to be unlinked in the populations Kanota x Marion and 'Aslak' x 'Matilda'[13,14]. On the consensus map, however, the

**Table 1 Regions of the oat genome containing major QTLs and/or candidate genes.**

| Chromosome | Region[a] (Mbp) | QTLs[b] | Alleles[d] | Frequency[e] | Candidates[f] |
|---|---|---|---|---|---|
| Chr1A | 359–419 | GRT.ShHf.375[c], GRT.GoHf.361[c] | GRT.Go = 1.1, GRT.Sh = 1.2 | GRT: 2/3 | |
| Chr1AC | 435–476 | GRT.GoHf.457[c] | GRT.Go = 1.2 | GRT: 2/3 | |
| Chr1C; 1CA | 44–427; 424–427 | PRT.ShHf.377, PRT.ShHf.425[c] | PRT.Hf = 0.6 | PRT: 2/4 | |
| Chr2A | 336–357 | GRT.GoHf.340 | GRT.Hf = 1.6 | GRT: 1/3 | |
| Chr2C | 518–522 | KWT.TxHd.519 | KWT.Tx = 2.5 | KWT: 1/4 | |
| Chr2D | 150–173 | KWT.GoHf.156 | KWT.Go = 1.0 | KWT: 2/4 | |
| | 167–171 | LDG.TxHd.171 | LDG.Hd = 0.4 | LDG: 1/5 | |
| | 299–431 | GRT.ShHf.306 | GRT.Sh = 1.3 | GRT: 2/3 | |
| Chr4A | 274–318 | HED.TxHd.318[c], HGT.TxHd.283, LDG.TxHd.288 | HED.Tx = 2.6, HGT.Hd = 6.6, LDG.Hd = 0.5 | HED: 1/5 + 3 HGT: 1/5 + 4 LDG: 1/5 | Vrn1.292 |
| Chr4C | 2–42 | OIL.DaEx.16, OIL.GoHf.25 | OIL.Da = 1.2, OIL.Hf = 0.8 | OIL: 2/4 + 5 | ACC.24 |
| Chr4D | 268–289 | LDG.TxHd.272, HGT.TxHd.278, HED.TxHd.278 | LDG.Hd = 0.3, HGT.Tx = 10, HED.Tx = 7 | LDG: 1/5 HGT: 1/5 + 2 HED: 1/5 = 7 | Vrn1.277 |
| | 407–415 | TM: (YLD, TWT, THN, PLP, KWT, GRT) all at 411 Mbp | YLD.Ma = 126, TWT.Te = 7, THN.Te = 31, PLP.Ma = 31, KWT.Ma = 9, GRT.Te = 24 | YLD: 1/3 TWT: 1/3 THN: 1/1 PLP: 1/1 KWT: 1/4 GRT: 1/3 | N1.411 |
| Chr5D | 74–261 | OIL.TeMa.93 | OIL.Ma = 0.5 | OIL: 1/4 + 1 | |
| Chr6A | 314–330 | OIL.DaEx.314, BGL.GoHf.325 | OIL.Da = 1.2, BGL.Hf = 0.5 | OIL: 1/4 BGL: 2/3 + 1 | CslF11.298 |
| | 394–427 | BGL.GoHf.413, BGL.ShHf.415 | BGL.Hf = 0.4 (Sh)–0.8 (Go) | BGL: 2/3 + 3 | CslF9.416 |
| | 411–445 | OIL.GoHf.413, OIL.ShHf.415, OIL.DaEx.419 | OIL.Hf = 1.6 (Go) to 1.8 (Sh) OIL.Da = 1.7 | OIL: 3/4 + 4 | ACC.416 |
| | 398–410 | KWT.TxHd.405 | KWT.Tx = 2.5 | KWT: 2/4 | |
| Chr6C | 503–541 | UDH.ShHf.318, GRT.ShHf.541 | UDH.Hf = 0.7, GRT.Sh = 1.3 | UDH: 1/3 GRT: 1/3 | |
| Chr6D | 36–132 | LDG.DaEx.79 | LDG.Ex = 2.3 | LDG: 1/5 | |
| | 234–270 | HGT.DaEx.244, HED.DaEx.266 | HGT.Ex = 5.0, HED.Ex = 2.2 | HGT: 2/5 + 7 HED: 1/5 | |
| Chr7A | 60–72 | HED.DaEx.62[c], HED.TxHd.66, HGT.TxHd.66, KWT.TxHd.66 | HED.Hd = 6, HED.Ex = 3, HGT.Hd = 5, KWT.Tx = 3 | HED: 3/5 + 8 HGT: 2/5 + 2 KWT: 1/4 | VRN3.69, CO1.106 |
| | 131–432 | BGL.ShHf.426 | BGL.Hf = 0.4 | BGL: 1/3 + 2 | CSLF6.399 |
| Chr7D | 1–6 | OIL.DaEx.3 | OIL.Da = 1.1 | OIL: 1/4 + 1 | |
| | 3–480 | HED.TxHd.10 | HED.Hd = 6 | HED: 3/5 + 4 | VRN3.469, CO1.24 |
| | 427–464 | OIL.DaEx.459 | OIL.Da = 1.2 | OIL: 1/4 + 1 | |
| | 467–486 | PRT.GoHf.472, HGT.ShHf.486 | PRT.Go = 0.7, HGT.Hf = 3 | PRT: 2/4 HGT: 1/4 + 1 | |

[a]The region is the widest peak where the QTL is located based on a LOD fall-off of 20%. Regions are combined where there is reason to hypothesize that the QTLs could be related. Multiple regions are shown if the peak span differs substantially by population.
[b]The QTL name refers to Supplementary Data 5, omitting the term MEANS and the chromosome identifier (i.e., just the trait, population, and peak Mbp). All QTLs are based on the GBSi-imputed marker data and are significant at LOD ≥5 (TS >23) corresponding to a genome-wide error rate of ~0.001 unless indicated. This method of error control compensates for different population sizes and the presence of a large number of correlated QTL tests across the genome.
[c]Indicates that the QTL met a threshold of LOD ≥4 (TS >18.5) but is included due to the relevance of the chromosome region.
[d]QTL alleles are identified by the parent producing the high numeric value of the trait followed by the additive substitution effect of homozygous substitution. This is preceded by the trait name only where it is ambiguous.
[e]Frequency of detection in populations where the trait was measured (including those detected at LOD 3 and in single environments from ALL-QTL, GBSi, Supplementary Data 5) + the number of historical populations where the QTL was detected (from Supplementary Data 7).
[f]The candidate gene is followed by a position in Mbp based on information in Supplementary Data 6.

RFLP-based *ACC* loci are located on Mrg11, while other, closely linked markers from LG11 are found on Mrg05. Since two of the three *ACC* candidate gene loci examined in this study are physically located on chr4C and 6A, and since these chromosomes correspond to linkage groups Mrg11 and Mrg05, it is likely that the two RFLP-based *ACC* loci found previously match the genes on these two chromosomes, suggesting the possibility that the *ACC* gene on chr5D is either inactive or is not polymorphic.

Of the three loci for the heading-related gene *VRN*1, Vrn1_chr4D_277Mb is associated with the most QTLs from both this and previous studies; however, all three *VRN*1 loci have been associated with QTLs for vernalization response[15–17]. Of the three *VRN*3 loci, Vrn3_chr7A_69Mb, and Vrn3_chr7D_469Mb are associated with the most heading-related QTLs. Of the three loci for *CO1*, CO1_chr2D_22Mb was not associated with any QTLs, historical or otherwise. The CO1_chr7A_106Mb gene is close to Vrn3_chr7A_69Mb, but it falls outside and further away from the QTLs detected in this region. While the *VRN*3 and *CO1* genes on chr7D are further apart physically than those on chr7A, linkage groups corresponding to chr7D on genetic maps can show greatly reduced recombination. As a result, many of the historical QTLs on chr7D are associated with both genes. As an example of this, marker acor221 was located 10 cM away from a gene for daylength insensitivity (*Di*1) in the cross 'Premier' x

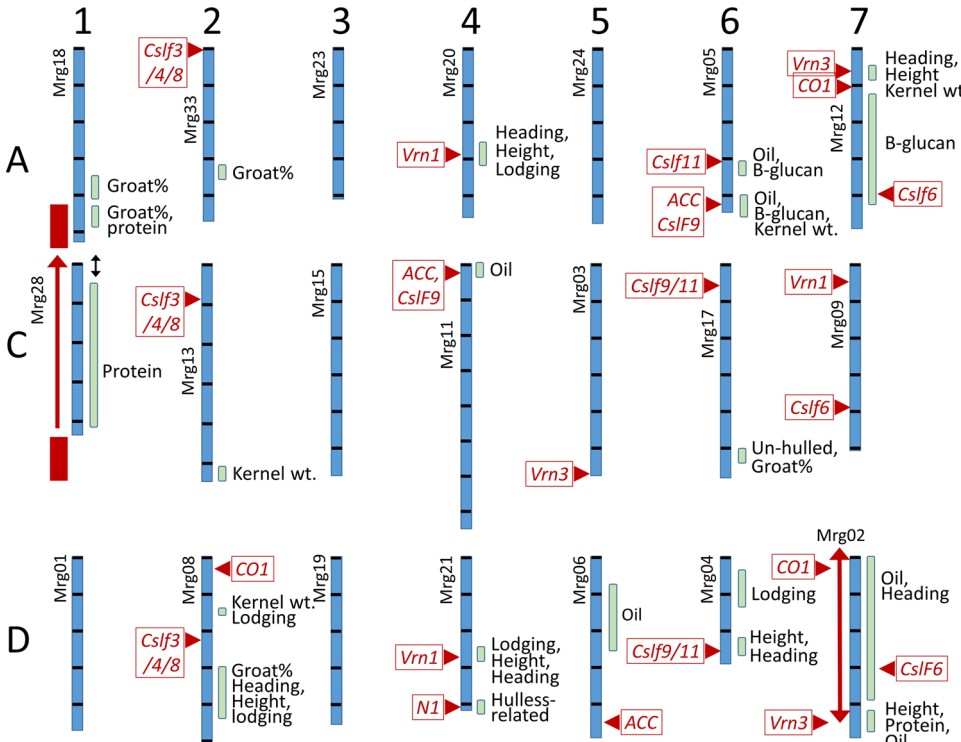

**Fig. 2 Positions of major QTL on 21 oat chromosomes relative to candidate genes.** Positions and 20% fall-off intervals of major QTLs (LOD >5) affecting trait means across environments in five oat populations (green bars) are shown in relation to positions of potential candidate genes (red boxes with triangles) relative to the Sang reference genome. Black bars show 100 Mbp intervals with base 1 at the top of each chromosome. Consensus linkage group names ('Mrg' identifiers) were identified based on a majority of corresponding markers. QTLs are defined by green bars indicating regions where the test statistic remains within 20% of the LOD peak. Further details and effects of QTL alleles are shown in Table 1 and Supplementary Data 5. Translocation of the region on chr1C (red block) to 1A is indicated by a red arrow (see also Fig. 1) suggesting that the QTLs in these two regions may be the same due to pseudo-linkage between chromosomes (double-headed arrow). Recombination suppression on chr7D, potentially caused by an inversion (red double-headed arrow, see also Fig. 1) results in long parental haplotypes and QTLs that cannot be localized in the cross TxHd.

**Table 2 Probability of observing at least *K* QTL co-located with candidate genes.**

| Trait | QTLs | Genome coverage (*p*)[a] | N. candidate genes (*M*)[b] | Observed co-location | *P(K≥1)* | *P(K≥2)* | *P(K≥3)* | *P(K≥4)* |
|---|---|---|---|---|---|---|---|---|
| OIL | 6 | 0.029 | 3 | 2 | 0.085[c] | **0.0025**[d] | 0.000024 | NA |
|  |  |  |  |  | 0.101[c] | **0.0031**[d] | 0.000028 |  |
| BGL | 3 | 0.034 | 11 | 3 | 0.316 | 0.0519 | **0.0053** | 0.00036 |
|  |  |  |  |  | 0.243 | 0.0278 | **0.0018** | 0.00007 |
| HED | 5 | 0.049 | 9 | 4 | 0.364 | 0.0687 | 0.0079 | **0.00059** |
|  |  |  |  |  | 0.450 | 0.1067 | 0.0150 | **0.00133** |

[a]Calculated by adding the cumulative QTL intervals for a given trait from Table 1, divided by the cumulative pseudomolecule size of 10,272 Mbp in the Sang genome. Each QTL interval is increased by 71 Mbp when testing against historical QTL positions.
[b]Based on the total number of candidate genes examined, from Supplementary Data 6.
[c]The first *P* value is predicted by the binomial formula; the second is based on $1 \times 10^8$ bootstrap simulations where random candidate genes were selected based on the observed gene density in the Sang genome and positions were restricted to one gene per 100 Mbp.
[d]Probability for the observed overlap is shown in bold.

'OA906-1-16'[18]. It was subsequently mapped to chr7D in the Kanota x Ogle population[17,19]. Instead of being found in the region associated with *CO1* in the current study, however, it was found in the region associated with *VRN3*. Thus, in populations where there is evidence of greatly reduced recombination, markers may not be able to distinguish between the effects of *CO1* or *VRN3* unless they are very tightly linked.

Height has often been associated with heading date, and this is reflected in the results here for QTLs on chromosomes 4A, 4D, 6D, and 7 A, but not 7D. While candidate genes for height were not examined, the height QTLs on chr6D are associated with the location of the dwarfing gene *Dw*6 on Mrg04 of the consensus map[20,21].

## Discussion

Here we report reference-based marker and quantitative trait analyses of five RIL populations representing nine diverse oat varieties. These analyses provide a transect of key QTLs encountered by oat breeders, the magnitude of their effects, their potential causes, and their context within the complex recombination landscape of hexaploid oat. We also provide evidence that this recombination landscape is affected by major heterogeneous structural chromosome rearrangements, such as a chr1C → chr1A translocation and a large pericentric inversion or other rearrangements on chr7D. Since reference-based oat genome analysis has only recently become feasible, our detailed

Supplementary Material will allow much deeper dives into the recombination landscape of oat and into the many additional QTLs with smaller or environment-specific effects that may be validated by future research.

During the development of the oat consensus map based on twelve RIL populations[3], we noted that maps from individual crosses sometimes differed from a representative consensus. Hence, we speculated about the presence and locations of chromosome rearrangements, including on consensus linkage groups corresponding to chr1A, chr1C, and chr7D[3]. However, due to the small population sizes and the lack of a reference genome, we could not conclusively identify the locations or causes of these differences.

The presence of a chr1C → chr1A translocation is well-known and documented[10]. However, the translocation was assumed to be reciprocal[22], which is a more common event. Our previous attempts at developing accurate linkage maps in populations containing this translocation were unsuccessful[23,24]. In the current work, we demonstrate that this translocation is highly unbalanced within the Sang genome, with a 106 Mbp chr1C → chr1A fragment vs. a 5 Mbp chr1C → chr1A fragment. By comparing recombination signals directly to the Sang reference genome, we provide evidence for the location and influence of this translocation in two RIL populations.

An obvious implication of rearrangements is the suppression of recombination, and, in the case of the chr1C → chr1A translocation, the presence of pseudo-linkage between two chromosomes. This causes an over-abundance of long parental haplotypes in populations derived from an $F_1$ in which an inversion or translocation is heterozygous. In the current populations, these two rearrangements result in long QTL intervals, with the result that accurate localization of a QTL or any map-based cloning is impossible. Thus, for example, the QTL for heading date in TxHd on chr7D may be caused by allelic variation at the *CO*1 locus, but it may also be caused by gene differences at any location across most of this chromosome, including the *VRN*3 locus near the opposite end of the chromosome. Since contrasting QTL alleles may be fully associated with the inversion/rearrangement, it may not be possible to localize the QTL in a cross where the inversion does not segregate. Thus, other functional genomics methods may be required to establish a causal mechanism for this important QTL.

Similarly, long haplotypes that couple physically distant genes can also dominate mixed diversity populations, affecting genome-wide association studies. This appears to have been the case in the oat CORE population[25], where we noted that chr1A, chr1C, and chr7D contain loci that are highly correlated with population structure[25] that span most or all of these respective chromosomes. Chr7D and chr1C also showed signatures of breeding history[2] which included a relatively high number of haplotype associations with heading date. Pseudo-linkage between chromosomes chr1C and chr1A and major QTL affecting winter-hardiness (winter field survival and crown freeze tolerance) is known[26], as is a clear geographic boundary at the southern edge of the Anatolian Plateau separating spring-habit landraces of *A. sativa* on the plateau carrying the chr1C → chr1A translocation from winter-habit landraces of *A. byzantina* in the lowlands that do not carry the translocation[10]. In addition to winter survival and geographic adaptation, loci in chr1C → chr1A translocated genomic regions are associated with many QTLs for disease resistance, kernel quality traits, plant architecture traits, heading date, and yield, as summarized in Supplementary Data 7.

In the populations studied here, the presence of rearrangements on chr1A, chr1C, and chr7D will make it difficult to identify recombination between QTLs for groat% and protein on chr1A and chr1C, and also in identifying recombination between QTLs for heading date, oil, and β-glucan on chr7D. These associations probably account for some of the unfavorable trait correlations that we have observed in the GoHf and ShHf populations[6]. Nevertheless, we also identified non-linked QTLs for most of these traits, which would account for our ability to select superior breeding lines from these populations[6].

The phenomenon of pseudo-linkage may have coupled genes that jointly confer adaptation to specific environments. This appears to be the case with flowering time and winter-hardiness, as conferred by genes associated with the 1A–1C translocation[22]. In many cases, this may provide an advantage to plant breeders. Breeders can make crosses that recombine other traits regulated by non-affected chromosomes, while recovering blocks of non-recombining adaptive genes. Often, crosses are made between spring oat varieties adapted to temperate environments and southern or winter-adapted varieties. The latter often carry new or more durable crown rust resistance, thought to be caused by increased prevalence of the pathogen in warmer regions where oat is cropped for extended seasons. However, where these traits are controlled by regions such as those on chr1A, chr1C, or chr7D, such introgression may not be successful, or it may introduce nonadaptive traits to the target environment. Going forward, oat breeders will be able to use the Sang reference genome and results such as those reported here to anticipate or avoid attempts to select intractable results.

The fact that all QTL were not found in all populations is a reminder of the need to examine multiple biparental populations when searching for key QTLs and/or selecting breeding targets, and a reason for including five diverse populations in this study. An example is the QTL for oil on chr4C, which we found only in DaEx and GoHf, or the QTL for β-glucan on chr7A, which we found only in ShHf—despite relaxing the criteria in the other populations (Table 1). Both of these QTL were highly significant, with additional support from candidate genes and from several historical QTL studies, but neither were found in both GoHf and ShHf populations, despite the common parent HiFi. This suggests that Sherwood, a selection from the cross 'AC Aylmer' x Goslin, contains AC Aylmer alleles at both of these two QTL: the oil allele on chr4C being similar to HiFi, and the β-glucan allele on chr7A showing a contrast with HiFi that was absent in Goslin. In contrast, we found QTLs for heading date on chromosomes 7A and 7D in three of the five populations, and also in many historical studies. This suggests that alleles at these heading-related loci are more likely to differ between parents of QTL studies. This may be due partially to how plant breeding progresses, and how crosses for QTL studies are chosen. While crosses for breeding purposes are often made between highly adapted lines with very subtle differences, crosses for QTL studies are often made between lines with contrasting adaptation: this is done both to increase the probability of QTL discovery as well as to generate the possibility of recombining alleles from parents that would not normally be used in a breeding program. A conclusion from this might be that molecular tools to assist in recombining QTL are most effective when used in wide crosses, where the effects of adaptive loci such as heading date can be separated more objectively from those affecting traits such as seed quality.

The discovery of a high rate of correspondence between major QTL and candidate genes for oil, β-glucan, and heading date may open up new possibilities and justifications for gene cloning in oat. It may also provide justification for more target-focussed methods of genomic selection. The candidate genes analysed in this study have shown evidence for involvement in the phenotypes of major economic importance in other crops; thus, their association with quantitative traits in oat may draw interest from

specialists in molecular biology and plant physiology who may be interested in working with a different plant model, or in advancing the science of a key functional food crop like oat.

The *ACC* gene encodes the first enzyme involved in the pathway that catalyzes acetyl-CoA to form malonyl-CoA. Because it is a rate-limiting step in the formation of fatty acids, it has been implicated in many studies as a potential modulator of oil accumulation. The association between *ACC* and groat oil was first noted in two crosses[13], both with a major oil QTL that corresponds to our oil QTLs on the distal end of chr6A.

The *VRN*1 gene is the grass ortholog of the *Arabidopsis* meristem identity gene *APETALA*1 (*AP*1) and is upregulated by vernalization in winter wheat[27], while *VRN*3 (=*FT*1) is involved in both the vernalization and photoperiodic pathways[28]. We have previously found associations between the location of *VRN*1 on chr4D and heading date, especially under late-planted conditions[17] and between *VRN*3 and heading date[17,25], especially under short-day conditions[19].

The *ClsF* genes belong to a large superfamily of cellulose synthesis genes[29]. The *ClsF* family, especially *ClsF*6, has previously been implicated in the formation of β-glucan in barley[30,31]. Recent work in oat[32] revealed a possible mechanism for the control of β-glucan by *CslF*6, whereby low-β-glucan varieties show the highest expression of the C-genome homeolog of *CslF*6, while high-β-glucan varieties, such as HiFi, have reduced C-genome expression. However, an association mapping study, as well as a QTL meta-analysis, did not reveal any QTLs at the C-genome location of *CslF*6 on linkage group Mrg09 (chr7C). Our current results confirm that there is no major QTL at this location, and instead show that the major β-glucan-increasing alleles contributed by HiFi are located near *CslF*9 on chr6A and *CslF*6 on chr7A. Further work on gene annotation and expression will be required to fully formulate a testable molecular mechanism for increasing β-glucan in oat. Due to the importance of this trait in human nutrition, this will continue to be a key focus of the oat research community.

Although this current report is necessarily limited in its scope and depth of functional gene analysis, it is striking that candidates from all of the tested gene families (i.e., *ClsF*, *Vrn*, *CO*1, and *ACC*) are associated with the positions of major QTLs in oat. In interpreting these associations, we expected to find three structural homeologous copies of each gene, but we did not necessarily expect that all three copies would produce functional RNAs and proteins, nor that all three would harbor QTL-related differences. It is interesting that, while some homeologous matches remain consistent on a given chromosome (e.g., *CslF*9 on chr6A, chr6C, and chr6D, or *CslF*3/4/8 on chr2A, chr2C, and chr2D), the positions of other candidate genes are not consistent among the oat sub-genomes. Homologs of most other genes appear to have moved to entirely different chromosomes. For example, the *VRN*1 genes are found in wheat and barley on orthologous chromosomes 5, but they appear to have moved to chr4A, chr4D, and chr7C in oat, despite the fact that the oat chromosomes have been named based on the ancestral identity to the Triticeae in their core centromere regions. This result is consistent with many earlier results[3,33,34] in suggesting that the hexaploid oat genome has undergone rapid and substantial chromosome restructuring.

Interestingly, the major QTLs associated with candidate genes are more numerous on chromosomes from the A and D genomes than those from the C genome. It is possible that there was a systematic loss of genes or suppression of gene expression in the C genome while the C and D genomes existed in a tetraploid before the hexaploid was formed by the addition of a new A genome[34]. Future work on transcriptome analysis and gene annotation will allow more detailed analysis and interpretation of this phenomenon.

It should be mentioned that some traits studied here (e.g., UDH and protein) have not been measured in very many studies. In addition, some of the older genetic maps lack enough information for proper comparisons to be made across populations, largely because of the choice of molecular marker type. Using bridging maps can also skew distances, and there are sometimes conflicts between the locations of markers on the original maps and the consensus map. Still, information from genetic maps can also be of great use, as was demonstrated by the unexpected finding of a link between the height QTL on chr6D and the location of the gene for *Dw*6. As none of the parents in the crosses used here are dwarfs, this could be an interesting gene to study. These caveats may explain the lack of historical QTLs corresponding to many of the major QTLs identified in this study; however, many of the minor or environment-specific QTL that we have identified for up to 14 different traits could also be novel regions worthy of further exploration, particularly those identified in both the GoHf and ShHf populations, or in the exceptionally large TxHd population.

Our team and many others are currently collaborating on the sequencing of over 24 reference genomes from different accessions of *A. sativa*. This, together with the ongoing development of new haplotype-based data imputation methods[35], is expected to revolutionize the ability to analyze population-based genotype data. This will also allow oat researchers to address the open question of the existence and location of further chromosome rearrangements among cultivated oats and their wild relatives. Knowledge of such rearrangements will allow further improvements in the interpretation and prediction of recombination events that affect oat breeding. Thus, while the chromosome rearrangements, major QTLs, and candidate genes presented in this work will provide a foundation for immediate applications, we look forward to future opportunities to fully integrate these results into a comprehensive gene and QTL atlas of cultivated oat.

## Methods

**Populations and traits**. The GoHf population consisted of 160 $F_{5:7}$ RIL lines, while the ShHf population included 215 $F_{4:6}$ RIL lines[6]. Of these, 159 and 207 lines were genotyped and analyzed in the current work. Previously reported phenotype data for GoHf and ShHf across up to eight site-year combinations (Table 3) is included in Supplementary Data 1. In 2012, subsets of 50 lines from each of GoHf and ShHf were grown;[6] thus, analyses of trait means for this year were performed separately, as abbreviated by MEAN12. The TeMa population consisted of 101 $F_{5:6}$ RILs[8], of which 70 were genotyped and analyzed in the current work. Previously reported phenotypic data for TeMa across up to 13 site-year combinations (Table 3) is included in Supplementary Data 1. The DaEx population included 146 $F_{5:8}$ RILs, all of which were reanalyzed in this work. Height, heading, and lodging were measured in two separately grown tests in Ottawa, and groat oil and protein were measured on pooled grain samples (Table 3 and Supplementary Data 1). The TxHd population consisted of 515 $F_{6:7}$ lines that were analyzed in this work. Phenotypes for 172 of these lines were collected in 2015 and also on previous families in the $F_{4:5}$ and $F_{5:6}$ generations in 2013 and 2015[9]. Here we report new data collected for heading date and plant height on an additional 343 lines beyond lines reported previously, dropping six lines with a missing marker or phenotype data from the 178 reported earlier; we also report new data on lodging for all progeny (Table 3 and Supplementary Data 1). Seed from parental lines are available as current commercial Canadian cultivars (Hidalgo, HiFi, and Marion) or canceled registrations (Goslin, Sherwood, Terra, Exeter) which may be available from companies listed at (https://inspection.canada.ca). Residual seeds of selected parental accessions and/or RIL progeny can be made available on a collaborative basis with a material transfer agreement.

**Marker analysis**. Marker analysis using genotyping-by-sequencing (GBS) was performed using established methods of complexity reduction by *PstI* / *MspI* digest followed by barcoding and high-throughput short-read sequencing[36]. For the GoHf, ShHf, and TeMa populations, GBS libraries were prepared by the Plateforme d'analyses génomiques, Institut de biologie intégrative et des systems (IBIS) group at Université Laval, and sequencing was performed using a NextSeq 500 sequencer (Illumina, San Diego, CA) with one mid-output sequencing lane per 96-sample library. For the DaEx population, raw data reported earlier[3] were fully reanalyzed as below. For the TxHd population, raw data reported earlier[9] were reanalyzed together with new data from 337 lines, for which GBS libraries were made and

**Table 3 Number of environments on which traits were measured for five RIL populations[a].**

| Trait | Description[b] | Goslin x HiFi (GoHf)[c] | Sherwood x HiFi (ShHf)[c] | Dal x Exeter (DaEx) | TX07cs-1948 x Hidalgo (TxHd) | Terra x Marion (TeMa) |
|---|---|---|---|---|---|---|
| BGF | Groat β -glucan (FIA analysis) | | | | | 5 |
| BGL | Groat β -glucan % (NIR analysis) | 8 | 7 | | | 6 |
| GRT | Groat percentage or milling yield (%) | 8 | 7 | | | 9 |
| HED | Days to 50% heading | 5 | 6 | 2 | 5 | 10 |
| HGT | Plant height at heading (cm) | 5 | 5 | 2 | 4 | 12 |
| KWT | 1000 Kernel weight (g) | 8 | 6 | | 1 | 9 |
| LDG | Lodging severity (%, 0–10, or 0–2) | 5 | 3 | 2 | 2 | 5 |
| OIL | Grain oil concentration (%) | 5 | 5 | 1 | | 13 |
| PLP | Plump kernels (%) | | | | | 2 |
| PRT | Grain protein concentration (%) | 5 | 5 | 1 | | 13 |
| THN | Thin kernels (%) | | | | | 6 |
| TWT | Test weight (kg hl$^{-1}$) | 7 | 6 | | | 5 |
| UDH | Un-dehulled grain (%) | 6 | 6 | | | 10 |
| YLD | Grain yield (kg/m$^3$) | 7 | 7 | | | 11 |

[a]The full list of environments, as well as all phenotypic data, were presented in Supplementary Data 1.
[b]Full details of populations and traits are found in De Koeyer, Tinker[8] for TeMa, in Sunstrum, Bekele[9] for TxHd, in Hizbai, Gardner[7] for DaEx, and in Yan, Frégeau-Reid[6] for GoHf and ShHf. Abbreviations used here are modified for consistency.
[c]Phenotypes for GoHf and ShHf populations were converted to LSMeans. For other populations, raw data were used.

HiSeq 2500 sequencing was performed by the University of Minnesota Genomics Centre.

The software package 'UNEAK'[37] was used to de-multiplex sequencing files and produce counts of all unique 64-base tags for each sample. Following this, Haplotag software[38] was run in production mode using input files that included a comprehensive set of 64-base tag-level haplotypes previously discovered in diverse populations[2]. Markers were filtered based on a maximum threshold of 50% missing genotypes, a minimum minor allele frequency (MAF) of 30%, and maximum heterozygosity of 10%. Genotypes were phased based on the genotypes of the mapping parents and ordered based on GBS marker locations in the Sang reference genome (Kamal et al., under review)[5] as described in Supplementary Data 2.

Because of a previously-known translocation between chromosomes 1C and 1A (previously named 7C and 17A, respectively)[10], GBS markers from these chromosomes were identified on the sequenced reference genomes of *A. eriantha* (C genome) and *A. atlantica* (A genome)[4] to confirm and locate this translocation in the Sang genome. Dot-plot comparisons among these genomes were then generated using Minimap2[39] implemented in the D-Genies package[40] to locate the position of the translocation breakpoint more precisely.

After filtering, marker data were sorted by reference genome position. Apparent outliers and reverse-phase markers were removed or corrected using simple heuristic algorithms (part of the GBSi program below) to count double crossovers. Data were then separated by chromosome and subjected to imputation using the program FSFHap[41], as implemented in the TASSEL 5.0 software package[42]. The FSFHap program is based on a Hidden Markov Model that adjusts genotypes to match predicted haplotypes and recombination breakpoints. For comparison, data were also imputed using the in-house program GBSi, which is based on heuristic algorithms designed to emulate decisions that would be used in manual curation. Details and parameters, as well as source code for the GBSi program, are available within Supplementary Data 2–4 for raw sorted data, FSFHap-imputed data, and GBSi-imputed data, respectively.

Following imputation, the average recombination distance between pairs of windows at tested coordinates throughout the genome was examined for each population. This was done by binning markers into 16 Mbp windows (8 Mbp right and left of tested coordinates) computed at sliding 1 or 10 Mbp coordinates, then computing the average recombination rate (r) between all pairs of markers in each pair of windows. For windows containing no markers, the markers immediately adjacent to the window were used to represent the window, such that all uniformly-spaced windows contained values. The resulting genome-by-genome recombination matrices were then visualized as heat maps with blended colors of yellow (r = 0) to teal (r = 0.2) to burgundy (r = 0.5).

**QTL analysis**. Traits were averaged over replications (where present), and QTL analysis was performed for each combination of population, trait, site, and year using the program MQTL[43]. Marker positions were converted to Mbp positions, such that the program accepted the values in place of cM positions. However, these values did not affect interval mapping, because the increment for interval mapping ('w') was set to 999, resulting in a single-marker analysis. Due to a previously identified strong single-marker association in the TeMa population, we included the marker avgbs_cluster_7805, at position 411,232,550 bp, as a covariate in all QTL analyses of this population. Analyses were also performed for each trait using means across all environments. For the GoHf and ShHf populations, the means from the year 2012 were analyzed separately from those of previous years. This was because selected subsets of only 50 lines per population were tested in this year,

which would have biased the use of overall means. The reduced population size in 2012 was a choice that was made to allow replicated tests in additional environments in order to facilitate better identification of breeding lines for use in future variety development[6].

The MQTL test statistic (TS) is a variance ratio, which is an approximation of the likelihood ratio[44]. For comparison to maximum likelihood methods for QTL analysis, which most often report the LOD value, TS was converted using the formula LOD = TS$(2\ ln(10))^{-1}$. Thresholds of TS (prior to converting to LOD) for declaring experiment-wise Type-I error rates were determined using 10,000 permutations for levels of p-values less than 5, 1, 0.5, and 0.1 percent. QTL peaks were inferred at marker positions where the TS exceeded a permuted threshold, and heuristic QTL intervals were formed by the region where the TS remained above 80% of the peak value. Once QTL intervals were declared, searches for additional QTLs on the same chromosome were made outside of a region defined by 75 Mbp above or below existing QTL regions.

**Candidate genes**. A search of the literature concerning genes in the Triticeae known to be associated with oil content, β-glucan, and heading date was conducted, and a non-exhaustive list of potential candidate genes was made. Gene sequences for the resulting list, which included Acetyl-CoA carboxylase (*ACC*), *VERNALIZATION*1 (*VRN*1), *VERNALIZATION*3 (*VRN*3), *CONSTANS*1 (*CO*1), and the β−1,3;1,4-glucan synthase-like gene series *CslF*3 through 12, were localized using a local instance of NCBI BLASTN (v.2.11) against the Sang reference genome. Initial BLAST parameters were set to evalue <0.005 and wordsize = 40, and the results were further filtered at evalue <1.00E-100, %identity >80, length >400, and score >200. Matching genome regions were manually curated to determine a composite start- and end-position for each gene. Associations between QTLs and candidate genes were proposed when a candidate gene fell within a heuristic QTL interval, as defined earlier.

**Historical QTL**. Historical QTL from previous literature associated with the candidate genes and QTL reported in this work were inspected. Regions of interest were first identified on the 2018 consensus map[2], either directly or through comparative mapping using the CMap tool in the GrainGenes database[45]. Relevant historical QTL in these regions, ±10 cM, were then compiled using information from published studies collected in GrainGenes. If none of the markers associated with candidate genes or QTL from the current study had been mapped directly on the 2018 consensus map, then other markers found in the same region of the Sang sequence that did have consensus map locations were used to locate the regions of interest instead.

**Statistics and reproducibility**. Statistical methods and population sizes within the above sections are summarized here as follows. Five RIL populations of sizes 160, 215, 146, 70, and 515 were used for QTL analysis. Populations were phenotyped for up 14 traits in field or greenhouse within variable numbers of environments (Table 3 and Supplementary Data 1). Most trials were conducted in non-replicated tests, and no specialized spatial designs were used to adjust observations. The use of non-replicated tests for QTL analysis is standard-practice, due to the fact that the genotypes are the experimental unit, and that genotypes are sampled more effectively by including more lines than by including smaller numbers of replicated lines (except in the unlikely case that all underlying QTL loci are present in all combinations for all traits)[46]. QTLs were identified in analyses of trait means as well as

for individual environments using single-marker tests. This was equivalent to performing a one-tail $t$-test of the average difference in phenotypes between two homozygous marker classes at each genetic marker. Due to the large number of tests performed, 10,000 permutations were run for each combination of trait and environment, whereby genotypes and phenotypes were reassociated randomly in each permutation to derive the distribution of the test statistic (TS) under the null hypothesis and to control genome-wide error rate at various levels of $p$.

To assess whether the co-location of candidate genes and QTL may have occurred randomly, the expectation ($P$) that $K$ or more out of a set of $M$ candidate genes distributed randomly in the genome would fall within a fixed set of QTL regions (defined by their heuristic 80% intervals) for a given trait that cumulatively occupied a proportion (p) of the genome was computed using the binomial formula (Eq. 1):

$$P(K \geq N) = \sum_{k=N}^{M} \binom{M}{k} p^k (1-p)^{M-k} \qquad (1)$$

This formula was used to generate expected distributions of K under the null hypothesis that candidate genes occurred randomly in relation to the QTL locations. The null hypothesis (no association) was tested using a one-tailed test and rejected at a value of $P$ corresponding to the observed number of overlaps.

The above approach may be limited by the following: (i) it is a prediction based on sampling with replacement, (ii) the real-life declaration of QTL/candidate coincidence is based on integers rounded to the nearest Mbp and (iii) gene density is not uniform across the genome. For this reason, a simple resampling algorithm (available by request) was written to derive bootstrap values for the coincidence of source QTLs and candidate genes, given a simulated genome made of 21 chromosomes divided into 1Mbp units, where candidate genes were placed with a probability that was based on measured gene density in the Sang genome, and where candidate genes were not allowed to be located within 100 Mbp of one another. These bootstrap values were compared to those estimated from the binomial formula for each set of tested parameters.

**Reporting summary**. Further information on research design is available in the Nature Research Reporting Summary linked to this article.

## Data availability

There are no restrictions on data availability. All primary DNA sequence reads from the pooled GBS libraries of five RIL populations and parental lines are available in the SRA division of NCBI (https://www.ncbi.nlm.nih.gov/bioproject/?term=PRJNA760785). Raw and imputed genotype calls and raw phenotype data are available in Supplementary Data 1 through 4. Pseudomolecules and gene annotations for the Sang v2 reference genome[5] are available for download from the GrainGenes database (https://wheat.pw.usda.gov/GG3/content/avena-sativa-sang-v1-dataset). Diploid genome sequences used for Supplementary Figs. 1 and 2 are available for download from: https://genomevolution.org/coge/GenomeInfo.pl?gid=53337 (Avena atlantica) and https://genomevolution.org/coge/GenomeInfo.pl?gid=53381 (Avena eriantha)[4]. Data for Fig. 1 and Supplementary Figs. 3 to 8 are available in Supplementary Data 4. Data for Fig. 2 is found within Table 1, which is based on Supplementary Data 5 and 6. Results of QTL analysis in Supplementary Data 5 are available in the GrainGenes[45] database for display as a track on the Sang reference genome (https://wheat.pw.usda.gov/jb?data=/ggds/oat-sang). The linked genotype and phenotype data are available from the T3/Oat database[47,48] (genotypes: https://oat.triticeaetoolbox.org/breeders_toolbox/protocol/52; phenotypes: https://oat.triticeaetoolbox.org/folder/4912).

## Code availability

Source code for the custom software GBSimpute used for data imputation is provided in Supplementary Data 4 as well as at the Zenodo archive (DOI: 10.5281/zenodo.5725714). The code is written in Free Pascal and can be compiled in the open-access multi-platform "Lazarus" environment. The compiled software generates a help file when executed without input parameters. Instructions, pseudocode, and a sample input file are also available in Supplementary Data 4. The corresponding author will provide a compiled version of the software by request.

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

## Acknowledgements

We are grateful to Brian Boyle and other staff at the Plateforme d'analyses génomiques, Institut de biologie intégrative et des systems (IBIS) group at Université Laval, staff at the University of Minnesota Genomics Centre, Julie Chapados, and other staff at the Ottawa Research and Development Centre, Agriculture and Agri-Food Canada, Molecular Technologies Centre, and Shiaoman Chao and staff at the USDA-ARS Fargo genotyping laboratories, all of whom contributed to GBS libraries and/or GBS sequencing of the populations. We also appreciate useful discussions with Asuka Itaya, Yung-Fen Huang, and Kyle Gardner, and we are grateful to Klaus Jakubinek, Steve Thomas, Brad de Haan, Allan Cummiskey, Dorothy Sibbitt, Frédérick Sunstrum, and Tina Wambach, who contributed technical assistance related to this work.

## Author contributions

N.A.T. conceived the reference- and candidate-based QTL study, performed marker analysis, data imputation, and QTL analysis, and led manuscript development; N.A.T. and W.Y. conceived individual population studies; W.Y. and C.P.W. led phenotypic data collection; N.S., M.S., and M.M. contributed early access to the Sang reference genome and related data, and coordinated joint analysis and interpretation of results; M.S. and T.L. developed gene models and assisted in interpretation of results; N.A.T., W.A.B., N.T.R., N.S., M.S., T.L., and M.M. conceptualized recombination profile analysis; C.P.W. interpreted historical QTL; E.N.J. contributed karyotypes based on C-banding; N.A.T., C.P.W., and W.A.B. interpreted QTL candidate genes; N.A.T., W.A.B., C.P.W., and E.N.J. interpreted implications of structural rearrangements; all authors contributed to manuscript development and approved the final manuscript submission.

## Competing interests

During manuscript preparation, Nick Sirijovski was an employee of a commercial enterprise Crop Tailor, a division of Lantmännen SE. At the time of manuscript revision, he became an employee of Oatly, SE. He was not involved in the design of the experiment and is not expected to benefit commercially from the publication of these results. The remaining authors declare no competing interests.
