## [Transparent Peer Review File · Communications Biology]

Reviewers' comments:

Reviewer #1 (Remarks to the Author):

This oat paper presents marker development and analysis of QTL affecting agronomic and nutritional traits in five oat populations. The paper relates QTL to candidate genes and to chromosome recombination and rearrangements. The paper provides practical information for the oat community and describes methods/applications which could be used more broadly for genetics research and crop improvement.

One of the three stated objectives (developing markers and expanding population size and phenotype data in TxHd, Lines 69-71) has a minimal place in results and discussion. Is new phenotype data reported? Consider including more direct results corresponding to this objective.

Results section, Marker analysis and imputation: consider reversing the order of these two paragraphs. This would place the general marker information first, and would provide better flow into the next section on chromosome recombination.

Lines 255-257: Why do many of the identified QTL not correspond to historical QTL? Were there historical QTL that were not identified in this study, and why might that be the case?

Fig. 1: Do these recombination landscapes correspond to previous cytogenetic studies of pairing configurations? It seems the Tx and Hd 7D chromosomes could facilitate pairing along a longer chromosome length (the diagonal striped section paired, and the speckled section unpaired) by having one chromosome reverse orientation.

Fig. 2: Is the protein QTL on 1C in the area of pseudo linkage (Fig. 1)? If not, why is the protein QTL not more localized?

Lines 319-320: The meiotic effects of translocation have been recognized and studied for decades (e.g. in oat: Shalev and Ladizinsky, 1976. The segregation pattern of a translocation quadrivalent, Chromosoma 57: 287-308). Consider revising to clarify what knowledge/application is novel.

Line 331: Could the QTL be analyzed in a population which is homozygous for the rearrangement (or for the normal chromosome structure)?

Below are suggested mechanical edits:

L33 – Two different types used for beta. Better to have consistent usage throughout. cf. lines 21, 51... vs. lines 245, 256...

L35 – As the first citation, this reference should be 1 (vs. 3). Check order of references throughout.

L41 – Are these just regular QTL studies?

L42 – from vs. form

L43 – have vs. has (two factors (plural) which cause challenges)

L54 – long days (no hyphen)

L55 – sensitive vs. sensitive

L60 – northern as lower case (cf. L54)

L85 – DaEx vs. DeEx

Table 1 header – GoHf vs. GsHf

L217 – space between chr1A and translocation

Fig. 1 legend – 4th line - unbalanced vs. unballanced; 6th line - pairing vs. paring; 7th and 9th lines - extra space after between?

Fig. 2 legend – 9th line - linkage vs. linked

L309 – comma vs. colon

L314 – previous vs. pevious

L316 – unbalanced vs. unballanced

Reviewer #2 (Remarks to the Author):

The manuscript "Genome analysis in *Avena sativa* reveals hidden barriers and opportunities for oat improvement" presents a comprehensive re-analysis of historical linkage mapping phenotypes using recently developed sequence-based genotypes. In addition, the authors place these findings relative to the newly developed reference genome sequence. The observations reported represent a significant advancement in oat QTL mapping and will likely be very useful to the oat breeding community. The statistical analyses were thorough and appropriate. The manuscript is consistently organized, making it easy to read. The language is accomplished and I found very few grammatical errors. Additional discussion comparing findings in the originally characterized 178 RILs of TxHd with those in the newly characterized 337 lines would be interesting. A few specific comments follow:

Line 42: "from" instead of "form"

Line 422: remove "in"

Table 2: Although clear in this context, consistently naming alleles with trait.parent would aid in using these results as points of comparison in future work.

Reviewer #3 (Remarks to the Author):

The authors had three main objectives: 1) To genotype at higher density five oat RIL populations; 2) To identify QTL with reference to the reference genome, and associate these QTL with candidate causal loci; 3) To identify any chromosomal anomalies by computing recombination "profiles" for each of the RIL populations.

To achieve the first aim, they used genotyping-by-sequencing (GBS) – compiling new and preexisting sequencing data for the five RIL populations. The authors then conducted QTL analyses for (at most) fifteen traits in five RIL populations using standard QTL mapping procedures. They subsequently assessed whether candidate genes associated with the phenotypic traits under study and/or previously identified QTL's colocalized with those identified in the present study, and found some evidence for this. However, without a more sophisticated statistical analysis, we cannot know whether or not the number of candidate genes associated with identified QTLs exceeded that expected by mere chance. Specifically, given the number of candidate genes and the genomic area occupied by identified QTL, and assuming the genes and QTLs are uniformly distributed across the oat genome, we would expect a certain number of genes to land in the QTL regions by chance. In addition, this analysis is conducted by grouping all candidate genes – presumably identified in a trait-specific manner – and QTLs together, which should artificially inflate the probability of colocalization.

In addition, despite the fact that the QTL analyses were conducted in five distinct populations, the differences in mapping results across populations were not explicitly discussed in the main text. (Although, they were summarized in a table in the main text.) Perhaps a figure comparing the QTL mapping results across populations for a subset of traits could address this. In addition, it would be interesting to note, how often did QTL replicate across populations? If not, is this due to lower power in a subset of the populations? I think that these questions are well within the scope of the study, and their answers have practical consequences for oat breeding programs.

To address the third aim, the authors used both genotype data and dot-plots derived from genome alignments to refine the breakpoint positions of a known translocation, Chr1C -> Chr1A. Pairwise recombination rates among all genomic regions were used to identify additional chromosomal

abnormalities. However, the authors primarily discussed recombination patterns in the region of this previously known translocation. In addition, the authors hypothesized that a chromosomal inversion suppressed recombination on Chr7D in the TxHd cross. In both cases, the authors provide models for hypothetical meiotic pairings that would lead to the observed recombination patterns. While these models are certainly plausible, I wonder if it would be possible to simulate realizations of the recombination matrices for comparison to the observed values. In other words, are the observed recombination matrices entirely consistent with the hypothesized models – namely for the more complex Chr1C -> Chr1A translocation?

The results of the third aim are of great consequence to oat breeders, as they represent a potential barrier to gene mapping and crop improvement efforts. While the authors speculate as to the adaptive consequences of these rearrangements, this is beyond the scope of their study.

The results of this study, are, in general, likely of great interest to oat breeders. However, due to the limited scope of the analyses, this paper is likely to garner little interest from those outside of oat breeding. In particular, the authors do not provide a compelling comparison of QTL results across the RIL populations – which would be of interest to those interested in the replicability of complex trait architecture across subpopulations. In addition, without further statistical analyses, it is difficult to interpret the candidate gene analyses. Further, while the localization of chromosomal rearrangements in oat is perhaps of wider interest, due to the study design, the authors cannot do much beyond identifying these rearrangements.

Below are some larger scale comments, that may partially repeat what is stated above:

(1) For the candidate gene analysis, you identify a set of genes that are associated with the traits of interest. You then ask which of the genes fall near an identified QTL. My question is: Given the number of genes on your list and the total size of the regions identified as QTLs, how many genes would you expect by chance to find in the QTL intervals if both were uniformly distributed throughout the genome? I think establishing this “null hypothesis” is important for interpreting the colocalization results.

(2) I have similar reservations about the colocalization of previous QTLs with those that you identified. How many would you expect to colocalize given the cumulative “area” that both sets occupy in the genome? Also, details about the previous QTL methods should be provided in the methods, not in the results. This methodology suffers from not specifying what is in fact “near”? The results are also fairly vague as to how close the known (“historical”) QTLs are to those identified in your study, e.g. in line 262-263, you write, “The two major HiFi QTL on chr6A appear to coincide with CsIF9 and 11.” What does “appear to coincide” mean? Unless this analysis can be made more precise, I do not think it should be included and/or you should only discuss those instances of colocalization that are strongly supported, i.e. within some maximum distance.

(3) In the introduction you give three aims. I would structure the methods and results in accordance with those aims (and their order) as much as possible.

(4) The discussion is an opportunity to integrate your results. As such, I would not separate the discussion into distinct sections corresponding to the results sections.

(5) The supplementary information is unwieldy. First, the slides in several supplementary sections should be combined in to a supplementary text with figures which both can be cited in the main text. Second, the excel sheets are difficult to navigate. The authors should also consider using plain text files to avoid error-prone excel altogether.

(6) This is perhaps only a matter of style: Where possible, I recommend switching to active voice. For example, the first sentence of your abstract could read: “We identified the positions and effects of

major..."

In the attachment, I provide comments on particular lines and paragraphs by section.

Itemized responses to three reviewers of manuscript “Genome analysis in *Avena sativa* reveals hidden breeding barriers and opportunities for oat improvement”

Notes: All reviewer comments are included in grey-highlighted text, and our responses are non-highlighted. We have added IDs to each reviewer comment so that responses can be cross-referenced. These are in the form RN.CN where RN refers to the reviewer number, and CN is a comment ID that we added where necessary. Comments that do not require a response are not numbered. Since reviewer 3 provided three versions of comments at different levels of detail (general + numbered + line-numbered), we have identified these as R3A, R3B, and R3C, respectively. Line numbers that refer to locations of revisions in the new manuscript all identified using {curly brackets}.

We have refrained from thanking reviewers for every individual comment, but please be assured that we are grateful for the exceptional in-depth reviews that were provided, and for the considerable time that reviewers spent with the goal of helping us to improve this manuscript. We hope that the revised manuscript will demonstrate that this has been the result.

Reviewer #1 (Remarks to the Author):

R1: This oat paper presents marker development and analysis of QTL affecting agronomic and nutritional traits in five oat populations. The paper relates QTL to candidate genes and to chromosome recombination and rearrangements. The paper provides practical information for the oat community and describes methods/applications which could be used more broadly for genetics research and crop improvement.

R1.1: One of the three stated objectives (developing markers and expanding population size and phenotype data in TxHd, Lines 69-71) has a minimal place in results and discussion. Is new phenotype data reported? Consider including more direct results corresponding to this objective.

Indeed, TxHd is the only population where new phenotype data is reported other than some minor traits that were added to GoHf and ShHf. This is implied in the objectives, as stated above {83}, and also in the methods {105-107}. We have now added a header-note in the phenotype data for TxHd in Supplementary Data 1, such that the previously reported phenotype data can be distinguished from the new data collected in this study. We corrected an error in the methods to identify that the overlap between these studies was 172 lines and not 178 {104}. This was a result of dropping some lines with missing marker or phenotype data, as we now note. We also note {107} that all lodging data in this population is new.

While a full comparative analysis on the effect of increasing the population size of TxHd on QTL for flowering time and heading date was not an objective of this study, we agree that it could be of interest to many readers, and possibly it can expand this interest beyond the oat community. Therefore we have added an additional table in Supplementary Data 5 showing a full comparison of the within-environment QTL results for height and heading date for the complete (current) 515-RIL population compared to the parallel analysis from the previous report by Sunstrum et al. (2019). A summary of the results is given on lines {362-371}: Essentially, all QTL except one were re-detected in the new larger population. The QTL were not at new locations, but they were detected with much higher significance and in more of the environments.

R1.2: Results section, Marker analysis and imputation: consider reversing the order of these two paragraphs. This would place the general marker information first, and would provide better flow into the next section on chromosome recombination.

We have reversed the first two paragraphs of the results on marker analysis and imputation, as suggested {255-283}. This reversal required a “see below” statement {264} where the translocation is now mentioned ahead of its result, but we agree that this allows us to begin with more general information.

R1.3: Lines 255-257: Why do many of the identified QTL not correspond to historical QTL? Were there historical QTL that were not identified in this study, and why might that be the case?

We have refocused the introductory paragraph in the QTL results {393} by mentioning first that QTL co-location was strong for the traits height, heading, and B-glucan, and other additional information on historical QTL overlap {393-407}. This includes suggestions on why historical QTL may not always coincide with current QTL {401-408} which is expanded further in the discussion {beginning on line 550}. In short, the overlap among QTL for some traits is low because they are measured only in a small number of populations, or because they are affected by a larger number of genes, but also because we don't expect to find the same QTL in every bi-parental population, and some of the traits studied here have not been analyzed in previous literature, or vice-versa.

See also the response to **R3A.2**.

R1.4: Fig. 1: Do these recombination landscapes correspond to previous cytogenetic studies of pairing configurations? It seems the Tx and Hd 7D chromosomes could facilitate pairing along a longer chromosome length (the diagonal striped section paired, and the speckled section unpaired) by having one chromosome reverse orientation.

This is an excellent point. We are not sure why the recombination suppression occupies such a large region of 7D, and the reviewer is correct that a reversed pairing might occur if the rearrangement was a simple large inversion. We have now acknowledged this concern, provided further speculation that a more complex set of multiple inversions or other rearrangements could exist, and explained that this puzzle can probably not be solved until additional reference genomes are fully sequenced {see Abstract, line 31, Results lines 325-329, Figure 1, and Discussion line 482}.

R1.5: Fig. 2: Is the protein QTL on 1C in the area of pseudo linkage (Fig. 1)? If not, why is the protein QTL not more localized?

The protein QTL on chr1C occupies the full region of pseudo linkage, and is detected on both chr1C and the 1C→1A translocation. There was an error in Figure 2 where it was not explicitly shown on both regions, and this has now been corrected.

R1.6: Lines 319-320: The meiotic effects of translocation have been recognized and studied for decades (e.g. in oat: Shalev and Ladizinsky. 1976. The segregation pattern of a translocation quadrivalent, Chromosoma 57: 287-308). Consider revising to clarify what knowledge/application is novel.

We had previously considered that the characterization of a highly non-balanced translocation was novel, to the extent that we still have not found a good diagram of this in the literature. This remains the case, but we have deleted the novelty statement {that would have followed line 501} from the revised manuscript because it is not critical to the impact of this study, and we don't wish to claim this uncertain novelty. (previously "*We did not find any previously published diagrams or models for the meiotic effects caused by unbalanced translocations. Thus, our diagram (Figure 1) and the associated empirical data may have broader applications to identify and characterize non-reciprocal translocations in other species.*")

R1.7: Line 331: Could the QTL be analyzed in a population which is homozygous for the rearrangement (or for the normal chromosome structure)?

This is a good suggestion, however the QTL may not differ between parents that share the same configuration; and if it did, we may not be certain that it is the same QTL. We have commented on this in an additional sentence at {512-515}: "*Since contrasting QTL alleles may be fully associated with the inversion/rearrangement, it may not be possible to localize the QTL in a cross where the inversion does not segregate.*"

R1.8: Below are suggested mechanical edits:

All of the below have been addressed as recommended.

L33 – Two different types used for beta. Better to have consistent usage throughout. cf. lines 21, 51... vs. lines 245, 256... (fixed)

L35 – As the first citation, this reference should be 1 (vs. 3). Check order of references throughout. *this was caused by an auto-referencing glitch: Word numbered the references in the figure caption boxes before the body of the text. Corrected now.

L41 – Are these just regular QTL studies? (yes- deleted the S)

L42 – from vs. form (fixed)

L43 – have vs. has (two factors (plural) which cause challenges) (fixed)

L54 – long days (no hyphen) (fixed)

L55 – sensitive vs. sensitise (fixed)

L60 – northern as lower case (cf. L54) (fixed)

L85 – DaEx vs. DeEx (fixed)

Table 1 header – GoHf vs. GsHf (fixed)

L217 – space between chr1A and translocation (fixed)

Fig. 1 legend – 4th line - unbalanced vs. unballanced; 6th line - pairing vs. paring; 7th and 9th lines - extra space after between? (fixed)

Fig. 2 legend – 9th line - linkage vs. linked (fixed)

L309 – comma vs. colon (fixed)

L314 – previous vs. pevious (fixed)

L316 – unbalanced vs. unballanced (fixed)

L380 – varieties vs. varaieties (fixed)

Reviewer #2 (Remarks to the Author):

R2: The manuscript "Genome analysis in *Avena sativa* reveals hidden barriers and opportunities for oat improvement" presents a comprehensive re-analysis of historical linkage mapping phenotypes using recently developed sequence-based genotypes. In addition, the authors place these findings relative to the newly developed reference genome sequence. The observations reported represent a significant advancement in oat QTL mapping and will likely be very useful to the oat breeding community. The statistical analyses were thorough and appropriate. The manuscript is consistently organized, making it easy to read. The language is accomplished and I found very few grammatical errors.

R2.1: Additional discussion comparing findings in the originally characterized 178 RILs of TxHd with those in the newly characterized 337 lines would be interesting.

We have now addressed this (see **R1.1**).

A few specific comments follow:

R2.2: Line 42: "from" instead of "form"

Corrected

R2.3: Line 422: remove "in"

Corrected

R2.4: Table 2: Although clear in this context, consistently naming alleles with trait.parent would aid in using these results as points of comparison in future work.

We agree with the reviewer that consistency and potential future reference is more important than saving a few lines of space in Table 2, therefore we have added the trait prefix to all alleles in the revised Table 2 {465}.

Reviewer #3 (Remarks to the Author):

R3A: The authors had three main objectives: 1) To genotype at higher density five oat RIL populations; 2) To identify QTL with reference to the reference genome, and associate these QTL with candidate causal loci; 3) To identify any chromosomal anomalies by computing recombination "profiles" for each of the RIL populations.

R3A: To achieve the first aim, they used genotyping-by-sequencing (GBS) – compiling new and preexisting sequencing data for the five RIL populations. The authors then conducted QTL analyses for (at most) fifteen traits in five RIL populations using standard QTL mapping procedures. They

subsequently assessed whether candidate genes associated with the phenotypic traits under study and/or previously identified QTL's colocalized with those identified in the present study, and found some evidence for this.

R3A.1: However, without a more sophisticated statistical analysis, we cannot know whether or not the number of candidate genes associated with identified QTLs exceeded that expected by mere chance. Specifically, given the number of candidate genes and the genomic area occupied by identified QTL, and assuming the genes and QTLs are uniformly distributed across the oat genome, we would expect a certain number of genes to land in the QTL regions by chance. In addition, this analysis is conducted by grouping all candidate genes – presumably identified in a trait-specific manner – and QTLs together, which should artificially inflate the probability of colocalization.

We acknowledge this concern, especially since some QTL intervals are very large due to putative chromosome rearrangements. To address this concern, we have computed binomial probabilities of observing K or more overlapping QTL-vs-candidate genes, based on cumulative genome coverages for a given trait vs. the number of candidate genes tested for that trait. We supplemented these calculations with a set of bootstrap simulations intended to approximate the biological model more accurately. These methods are described {232-249} and the resulting calculations are reported in a new Table 3 {468}. Both methods involve grouping all candidate genes and QTL together in a trait-specific manner, which was a concern of the reviewer. These computations show that the observed co-localizations of QTLs and tested candidate genes exceed the null hypothesis of no overlap at a highly significant level ($P < 0.005$) {384-386}. Nevertheless, this does not prove the role of any candidate genes, since the causal gene of a QTL may be a regulatory element or another gene that is associated with the hypothesized candidate. We have explained these results further on lines {384-386, 389-392, 409-410, 510-515}.

R3A.2: In addition, despite the fact that the QTL analyses were conducted in five distinct populations, the differences in mapping results across populations were not explicitly discussed in the main text. (Although, they were summarized in a table in the main text.) Perhaps a figure comparing the QTL mapping results across populations for a subset of traits could address this. In addition, it would be interesting to note, how often did QTL replicate across populations? If not, is this due to lower power in a subset of the populations? I think that these questions are well within the scope of the study, and their answers have practical consequences for oat breeding programs.

This question raises interesting points about the objectives and implications of QTL mapping, and we are grateful for the opportunity to explore this further through additional discussion in the manuscript.

First, we have taken the advice to add a quantification of QTL presence across populations and historical. We did this directly in Table 2 {465}, by adding an additional column "Frequency", which summarizes the number of populations where a QTL was found, the number where the QTL was tested, and the number of historical QTLs found near that location. We considered adding these numbers to Figure 2, but decided not to because (A) it would clutter the figure and detract from the intent to show positional information and (B) because it would artificially distort the perception of QTL importance, as explained below.

We have now provided an elaboration in the Discussion section regarding why we do not expect to find the same QTL in different populations {550-569}. We have emphasized {357, 550-551} that this was a reason for examining multiple bi-parental populations. We provide an interesting example of how key oil B-glucan QTL alleles can differ, even between two highly related and similar parents {552-559}. Thus, if a major QTL does not replicate across bi-parental populations, it does not diminish the relevance of a QTL for use in a breeding application. This is in contrast to GWAS populations (which were not examined in this work) where QTL presence is also an indication of QTL importance and allele frequency. Of course, how often a QTL is found is also affected by how often a trait was measured and, unfortunately, the frequency with which a trait is measured varies substantially, both in the current study and in historical ones. Thus, systematic conclusions based on QTL frequency would be highly confounded by these factors.

We have now have alerted the reader to the fact that many of the traits studied here were tested in only one or two populations {355-356}. We have also focused beginning of the results to emphasize that the traits oil, B-glucan, and heading date had the most repeatable QTLs, in part for this reason {393}

See also our response to **R1.3**.

R3A: To address the third aim, the authors used both genotype data and dot-plots derived from genome alignments to refine the breakpoint positions of a known translocation, Chr1C -> Chr1A. Pairwise recombination rates among all genomic regions were used to identify additional chromosomal abnormalities. However, the authors primarily discussed recombination patterns in the region of this previously known translocation. In addition, the authors hypothesized that a chromosomal inversion suppressed recombination on Chr7D in the TxHd cross. In both cases, the authors provide models for hypothetical meiotic pairings that would lead to the observed recombination patterns.

R3A.3: While these models are certainly plausible, I wonder if it would be possible to simulate realizations of the recombination matrices for comparison to the observed values. In other words, are the observed recombination matrices entirely consistent with the hypothesized models – namely for the more complex Chr1C -> Chr1A translocation?

We agree that a simulation of meiotic events could be interesting, however we feel that this would go beyond the scope of this study. It is already common knowledge that meiotic events in a translocation heterozygote can lead to pseudo-linkage, and a simulation would necessarily create arbitrary parameters that would only demonstrate this in a circular fashion. The simulation could therefore be useful only to try and model gamete frequencies and test for skewed segregation. We think this would distract from the focus of an already-complex manuscript. We have, however, added a note that segregation ratios in SHf are skewed based on a chi-square test, and alerted readers to the possibility of gamete selection, such that further work could be pursued at a future date {313-317}.

R3A: The results of the third aim are of great consequence to oat breeders, as they represent a potential barrier to gene mapping and crop improvement efforts. While the authors speculate as to the adaptive consequences of these rearrangements, this is beyond the scope of their study.

We agree that this is beyond the scope of this manuscript, but we hope that these results together with further work on the oat pan genome will provide a more robust arena for future experiments to test the adaptive significance of chromosome rearrangements in oat, and their potential use in crop development.

R3A: The results of this study, are, in general, likely of great interest to oat breeders. However, due to the limited scope of the analyses, this paper is likely to garner little interest from those outside of oat breeding.

This is probably true to an extent of all trait-based QTL studies. Nevertheless, we feel that the current study, which incorporates an analysis across five populations (more than most QTL studies), a systematic examination of relevant candidate genes, and a highly visual summary of results, may attract attention beyond the oat breeding community. Now that oat reference sequences are available and an oat pan genome is in progress, this study will provide a much-needed focus on positional information regarding QTLs and chromosome rearrangements, and a groundwork for evaluating decisions to pursue gene cloning experiments. Previously, these have been limiting factors in this interesting and important crop species, and we hope that this manuscript will help to expand the interest in oat research beyond its traditional players.

R3A.5: In particular, the authors do not provide a compelling comparison of QTL results across the RIL populations — which would be of interest to those interested in the replicability of complex trait architecture across subpopulations.

Please see response to **R3A.2**.

R3A.6: In addition, without further statistical analyses, it is difficult to interpret the candidate gene analyses.

We have addressed this (see **R3A.1**).

R3A.7: Further, while the localization of chromosomal rearrangements in oat is perhaps of wider interest, due to the study design, the authors cannot do much beyond identifying these rearrangements.

We agree that this is only a start to what we believe is an important theme in oat genomics: the effect and implication of structural chromosome rearrangements. However we feel we have taken the subject beyond the stage of identifying rearrangements. While additional chromosome rearrangements are definitely present (we know this already from our preliminary work on the oat pan genome), we show that the three chromosomes affected by the rearrangements that are discussed in this work have influence at the population level {519-522} and we provide discussion about the breeding implications of these events {530-536}.

Below are some larger scale comments, that may partially repeat what is stated above:

R3B.1: For the candidate gene analysis, you identify a set of genes that are associated with the traits of interest. You then ask which of the genes fall near an identified QTL. My question is: Given the number of genes on your list and the total size of the regions identified as QTLs, how many genes would you

expect by chance to find in the QTL intervals if both were uniformly distributed throughout the genome? I think establishing this “null hypothesis” is important for interpreting the colocalization results.

We have addressed this (see **R3A.1**). The results are very convincing that the co-locations of QTL and candidate genes are not random.

R3B.2: I have similar reservations about the colocalization of previous QTLs with those that you identified. How many would you expect to colocalize given the cumulative “area” that both sets occupy in the genome? Also, details about the previous QTL methods should be provided in the methods, not in the results. This methodology suffers from not specifying what is in fact “near”? The results are also fairly vague as to how close the known (“historical”) QTLs are to those identified in your study, e.g. in line 262-263, you write, “The two major HiFi QTL on chr6A appear to coincide with Cslf9 and 11.” What does “appear to coincide” mean? Unless this analysis can be made more precise, I do not think it should be included and/or you should only discuss those instances of colocalization that are strongly supported, i.e. within some maximum distance.

Details of the historical QTL comparisons are now provided in the methods {207-214}, including our definition of “near” (10 cM). The words “appear to” have been removed {418}, since QTL coincidence is now defined by the methods. The identification of historical QTL locations is, by necessity, a best effort. We cannot ignore it, and it is sometimes valuable, but every historical study has its own methods and statistical criteria, and some have reported very numerous QTL locations based on methods and data that we cannot re-evaluate. We considered the possibility of quantifying the probability of overlap between current and historical QTL, which would theoretically fit within the framework of the calculations and simulations that we performed for the candidate genes {232-249}. However, the search space of the historical QTL could not realistically be defined. For the candidate genes, we decided “how many candidates do we know of, where are they, and let’s see if they coincide with the QTL that we declared”. For the historical QTL, we would need to distinguish between bi-parental populations (where, by definition, QTL are not always present) and GWAS studies (where important QTL should be present, but where every study uses different significance criteria). We have explained and acknowledged this uncertainty (397-401), and also the fact that our search for historical QTL is not intended to be a statistics-based result, but rather a convenience to readers interested in a specific QTL {394-396}.

R3B.3: In the introduction you give three aims. I would structure the methods and results in accordance with those aims (and their order) as much as possible.

As recommended, these are now four objectives {82-88}. These objectives are necessarily overlapping and dependent on common methods. We think that the manuscript structure is already logical and adequate. We looked at the possibility of a fully parallel set of headings, and concluded that it would be repetitive and awkward.

R3B.4: The discussion is an opportunity to integrate your results. As such, I would not separate the discussion into distinct sections corresponding to the results sections.

We have taken this advice and removed the subject headings in the discussion and added some integrating narrative {e.g. 550-552, 570-572}.

R3B.5: The supplementary information is unwieldy. First, the slides in several supplementary sections should be combined in to a supplementary text with figures which both can be cited in the main text. Second, the excel sheets are difficult to navigate. The authors should also consider using plain text files to avoid error-prone excel altogether.

We have addressed this to follow the journal requirements. The supplemental figures which were previously in supplements 5 and 6 have now been combined in a single supplement where they contain captions and figure numbers that are referred to in the text. The remaining supplements all contain data sets, and they have now been renumbered as Supplementary Data 1 through 7. All data are formatted as Excel files because we find that most users prefer this, and because this is a suggested format in the author instructions provided by the journal. Unlike plain text files, this format allows aggregation of multiple data matrices, and we have also used this format to attach a “legend” to each supplement to aid in creating metadata when these files are absorbed into online data resources. We separated the three sets of genotype data (Supplementary Data 2, 3 and 4) mainly on the basis of file size, and the fact that only the GBSimpute (Supplementary Data 4) are likely to be useful to most readers. The remaining sets are separated based on how they are likely to be used.

R3B.6: This is perhaps only a matter of style: Where possible, I recommend switching to active voice. For example, the first sentence of your abstract could read: “We identified the positions and effects of major...”

We also prefer the active voice when discussing results, since it helps to invite the reader into the conversation. We have revised the discussion section in several places to make this more consistent {e.g. 447, 488, 518, 534, 535, 629}. However we decided to leave the abstract, methods, and most of the results in the passive voice. It would be a very major and unnecessary task to revise this consistently to the active voice, and there are probably an equal number of readers who prefer passive vs active voice for these sections, which are intended to be objective and impartial.

In the attachment, I provide comments on particular lines and paragraphs by section. (Attachment is included and itemized below)

Abstract:

R3C.1: The phrase “potentially explained by candidate loci” is a bit strange. Why not just say: “candidate loci Vrn1, Vrn3, and CO1” were in the four identified QTL regions.

We have rephrased these statements to say that QTL were “co-located with” candidate loci {24-26}. This avoids the term “potentially explained” but still allows us to emphasize the completeness with which our detected QTL showed co-location with the candidate genes, and to emphasize the new co-location analysis that has been added to the manuscript based (see R3A.1).

R3C.2: “The effect of this translocation” is too vague. Perhaps, “Based on a model of quadrivalent meiotic pairing, we predicted that this translocation would cause pseudo-linkage and recombination suppression.”

We have taken this suggestion {28-29}.

R3C.3: Switch use of passive voice: “We also detected the presence of a major heterogenous...”

We prefer to retain the passive voice in the abstract (see **R3B.6**)

Introduction:

R3C.4: Line 32: There is a typo, it should read: “focuses”.

Corrected

R3C.5: Line 40: This is not a contrasting statement and there are typos. Here is an alternate phrasing: “Previous QTL studies have not benefited from a complete annotated reference genome, and those published before 2016 suffered from a lack of high-density marker loci.”

We have taken this suggestion {50-52}.

R3C.6: Line 47-8: Put the concluding phrase in active voice.

We prefer to retain the passive voice in the introduction (see **R3B.6**)

R3C.7: Line 55-6: I think that “lacked” should be in the present tense.

We now begin the sentence with “previously” such that the past tense of “lacked” is appropriate {64-65}.

R3C.8: Line 57: Change to: “Two other populations which have high-quality phenotypic data...and Terra’ x ‘Marion’, abbreviated...”

We have taken this suggestion, except that the word “that” is more correct than “which” so we used that instead {66-67}.

R3C.9: Line 65: What does “wide quantitative trait variation” mean?

We changed “wide” to “high” {74}.

R3C.10: Line 69: Change to present tense: “The primary objectives of the current work are:”.

This is a style preference. We prefer to write the objectives in the past tense, which is when they were considered and when the experiments were designed.

R3C.11: Line 69-70: The first aim seems like two aims: (1) to develop high-density marker data and (2) to expand the population and gather additional phenotypic data. But, I’m not sure that (2) needs to be included.

We have re-numbered the objectives to recognize four objectives. Since another reviewer recommended an expansion of the results reported for TxHd (see **R1.1**), and we have taken that advice, including this as a separate objective now makes more sense {82-88}.

R3C.12: Line 73: Change “thereby” to “in order to”.

We have taken this suggestion {86}.

R3C.13: Line 71: Consider rephrasing: “to develop comparative reference-guided QTL...” as it is not immediately clear what comparative reference-guided means.

We omitted the word “comparative” {84}.

Methods:

R3C.14: Line 94: Switch to active voice: “We performed marker analysis using...”

We prefer to retain the passive voice in the methods (see **R3B.6**)

R3C.15: Line 105: Can you say something more informative of the “input files developed from diverse populations”. What role do these files play in the analysis?

We have clarified this to read “...using input files that included a comprehensive set of 64-base tag-level haplotypes previously discovered in diverse populations” {127-128}.

R3C.16: Lines 105-6: Does “minimum completeness” refer to percentage of missing genotypes? If so, please just write that.

Suggestion taken {129}.

R3C.17: Line 126: This should read: “...the average recombination distance between...”

Corrected {149}.

R3C.18: Lines 126-8: Use of “positions” is a bit vague here. If I understand correctly, for this analysis, you compute a pairwise recombination distance for all two 16Mbp windows. I do not understand how you integrate results across sliding windows. In general, I think this procedure could be clarified.

We revised “positions” to “windows at tested coordinates” and clarified that we tested “sliding 1 or 10 Mbp coordinates” {149-153}. It should now be clear.

R3C.19: Line 132: Consider using a color-blind friendly palette.

We have revised the heatmaps to use a color-blind friendly palette of yellow, teal, and burgundy {lines 155-157, Figure 1, and Supplementary Figures 3 and 4}. We have used an online tool (<https://www.color-blindness.com/coblis-color-blindness-simulator/>) to verify that the contrast in all of our figures is now adequate that they can be interpreted by all common types of color blind users. References to color names in the legends may be a slight challenge, as with many scientific images, but we think these are obvious enough that they can be inferred from the contexts of the figures. Also, most color blind users are aware of accessibility settings (e.g. in Windows 10) that automatically recolor the screen palette in a way that is more suitable for their particular condition.

R3C.20: Line 134: Is a “simple mean” simply the mean? If so, just write, “We performed a QTL analysis for each combination of population, trait, site, and year, with traits averaged over replications, using the program MQTL.” Also, get rid of backslashes.

We have omitted the term “simple” on both locations and replaced the backslashes with commas {160-161 and 170}.

R3C.21: Line 135-8: These seem like unnecessary details that could be included in the supplement.

We would prefer to leave this sentence in the primary methods. The instructions to authors from the journal state that there is no restriction on length of the methods, and recommends that all relevant protocols should be described here to ensure reproducibility {162-164}.

R3C.22: Line 138-9: I don’t understand this statement: “...the effect of the hullless trait was compensated for by setting...” Here is an alternative phrasing of these two sentences: “Due to a previously identified strong single marker association in the TeMA population, we included this marker (avgbs_cluster_7805, position 411,232,550 bp) as a covariate.”

We have taken this suggestion {164-166}.

R3C.23: Lines 141-43: Please clarify both of these statements. Why were the population sizes reduced in 2012?

We have addressed this {170-175} as follows: *“For the GoHf and SHf populations, the means from the year 2012 means were analyzed separately from those of previous years. This was because selected subsets of only 50 lines per population were tested in this year, which would have biased the use of overall means. The reduced population size in 2012 was a choice that was made to allow replicated tests in additional environments: a comprise to facilitate better identification of breeding lines for use in variety development”*

R3C.24: Line 145: Should read: “...to maximum likelihood methods for QTL analysis...” And, what is meant by “show advantage”? Does it mean to be higher-powered?

We have replace v d the text as suggested {177} and removed the phrase referring to whether the maximum likelihood method will show an advantage {178-179}. Our meaning was that linear models have equal power to maximum likelihood methods for single marker tests, and that we had chosen single marker tests because of the high density of markers. In such cases, interval mapping can actually reduce the power of QTL detection, due to the presence of correlated tests (assuming control of global error rate is identical). However this level of detail is not necessary here.

R3C.25: Lines 145-48: I’m a bit confused why you bother with the transformation to an approximate LOD ratio given that it is a simple transformation. And, why is it only an approximate LOD ratio?

The conversion is made because most readers in this subject area will be more familiar with LOD values than with likelihood rations. The conversion and the reason that it is an approximation are explained in the classical reference Haley and Knott, 1992 (Heredity:69, 315-324), which we have now cited {177}

R3C.26: Line 148: Change to: “...10,000 permutations for significance levels of p-values less than 5, 10, and 15 percent.” (P was not defined.) Is this referring to the LOD ratio?

We have revised and clarified this {180-181}: *“Thresholds of TS (prior to converting to LOD) for declaring experiment-wise Type-I error rates were determined using 10,000 permutations for levels of p-values less than 5, 1, and 0.5 percent”.*

R3C.27: Line 149: Are these “confidence intervals” in a strict (statistical) sense? Or, is this just a heuristic for delineating the QTL region.

These are heuristic intervals, since there are no known methods to accurately determine QTL confidence intervals in the presence of diverse sources of error. We have revised “confidence intervals” to “heuristic QTL intervals” {183}. We also replaced the term “confidence interval” with “heuristic interval” elsewhere in the manuscript {202, 234}.

R3C.28: Line 154-57: Change to: “We identified candidate genes associated with the phenotypes under study by...” And, explicitly state how you identified the genes. An “analysis of the literature concerning genes” is a bit vague. Then continue, “We identified the positions of these # candidate genes using NCBI BLASTN...”

As stated earlier, we prefer to maintain the passive voice in the methods. However, in response to this comment, we have acknowledged the informal level at which this “analysis of the literature” was conducted as follows: “A search of the literature concerning genes in the *Triticeae* known to be associated with oil content, β -glucan, and heading date was conducted, and a non-exhaustive list of potential candidate genes was made.” {190-191}

R3C.29: Line 158-9: Are the gene sequences for Acetyl-CoA, etc. an additional gene set, or a subset of the candidate genes?

We clarified that the listed genes comprised the complete set that was searched: “Gene sequences for the resulting list, which included...” {194-196}. We also revised the order of this paragraph to make it clearer {189-203}.

Results:

R3C.30: Line 166: Change to active voice: “Based on the locations..., we determined that...”

We have maintained the passive voice (**R3B.6**).

R3C.31: Line 169: Change to: “We observed a very small reciprocal translocation starting at 422 Mbp and spanning only 5 Mbp of the distal part of chr1C.”

We prefer the passive voice (**R3B.6**), but we removed the term “evidence for” as it was redundant: “A very small reciprocal translocation on chr1C was observed from 422 Mbp...” {273}.

R3C.32: Line 176: Change to: “..., but we kept...”

We kept the passive voice (**R3B.6**), but revised this to “...but the bp coordinates from the original Sang chromosomes were preserved for ease of reference.” To correct the awkwardness of the word “keeping”. {281-282}

R3C.33: Line 179: What is meant by “data that were sorted”?

We have clarified this in the revised sentence: “...for data were markers were sorted by position in the reference genome...” {255-256}. *Note that this paragraph containing this revision was moved to the top of the results, as recommended by Reviewer 1 (R1.2).

R3C.34: Paragraph starting at line 179: Given that you only use the GBSimputed data set, I think that you should either omit mention of the other data sets entirely or relegate the details to supplementary material. In the same vein, I think that you should only include the number of markers in the BSimputed data set in the main text, and/or relegate all such numbers to a table in the supplementary materials. In addition, I find the two sentences spanning lines 186-188 to be poorly written and quite vague. For example, what is meant by “algorithmic limitations”?

We debated the option to omit completely the data imputed by FSHap as well as the non-imputed data. However, this might require a separate manuscript to evaluate our GBSimpute method (as we have done within this work), substantially delaying the reporting of these results. We also considered using only the FSFHap method, but it did not provide results for the shorter chromosome fragments. Thus we include both methods to show that the established method (FSHap) gave very similar results to GBSimpute, but that it had limitations on particular oat chromosomes. All of the data are already relegated to supplementary material. This paragraph {255-270} is the only mention of the comparison between imputation methods and we believe it is necessary. However, to avoid confusion about the QTL results, we have now removed the supplemental QTL analyses for FSFHap and non-imputed data, except for the comparison in the “major QTL” table within supplementary Data 5 {268}. We have also removed the term “algorithmic limitations” as it was not necessary {263}.

R3C.35: Line 196: What are the “selected regions” referred to here?

We clarified this to read “...for chromosomes 1A and 1C of GoHf and ShHf...” {288}.

R3C.36: Lines 195-6: Is this also a sliding window analysis, as was described in the methods?

Yes, this is based on the methods that were described. We added the word “sliding” to clarify this {286}.

R3C.37: Line 211-12: Remove “While this model appears complex” and just describe what the model is.

We have removed this phrase. We now begin the explanation with: “A simplified summary of this model is....” {305}

R3C.38: Line 221: Change to: “...TxHd cross on chr7D. Specifically, we only observed recombination on the lower distal telomeric region, despite an abundance of markers throughout the chromosome.”

We have revised this, but kept the passive voice (**R3B.6**): “Specifically, no recombination was observed on this chromosome except on the distal (lower) telomeric region...” {319-320}

R3C.39: Line 222: Isn’t an inversion a “genetic mechanism of recombination suppression”? What other mechanisms of suppression did you have in mind?

We had in mind genes that might suppress pairing, but this seems unlikely and unnecessary and we have removed this point {320-322}. Note that we have also expanded this paragraph in response to Reviewer 1 (**R1.4**) to suggest the additional possibility of multiple inversions that would prevent the chromosomes from pairing in an inverted manner {322-329}.

R3C.40: Line 224: I’m not sure that “interpreted” is the right word. Perhaps “depicted” would be better here.

We revised the word interpreted to depicted, as suggested {325}.

R3C.41: Lines 225-6: Change to: "...are lacking. Thus, we cannot identify which parent of TxHd has..."

We have revised this section to: *Rearrangements in this region were not visible via C-banding (Figure S5) because diagnostic C-bands on chr7D are lacking. Currently, we have no preliminary data to interpret which parent of TxHd has the ancestral (non-inverted) configuration, and a full characterization of this phenomenon will only be possible once additional reference genomes are available.* {329-333}

R3C.42: Line 237-38: Remove the sentence beginning "While these formatted..."

We feel it is important to keep this sentence {345-347}. It is a rationale for the sentence that follows, and it is also a reminder to those readers who may be interested in detailed comparative QTL analyses that they may wish to explore the supplementary data.

R3C.43: Lines 238-239: Remove the "Thus" at the beginning of this sentence. In addition, "most significant" does not make sense, as significance does not have degree. Change to "largest LOD values exceeding 5". Change the sentence beginning "This level was..." to "This LOD threshold was found by a permutation analysis..."

The word "Thus" is retained (to follow from the previous statement that we preserved). However we have taken these suggestions and revised the sentence to: *Thus, we will focus our discussion on Table 2, which describes QTL regions based on analyses of trait means having LOD values that exceed 5. This LOD threshold was found by permutation analysis...* {37-349}

R3C.44: Line 242: Remove the beginning of this sentence so that it reads: "In Figure 2, we visualize the major QTLs in relation to the set of candidate genes".

We have revised this to: *In Figure 2, the positions of these major QTL are visualized in relation to candidate genes on a scaled version of the Sang reference genome.* {351-353}

R3C.45: Line 245: What is meant by "candidate gene analysis"? Simply what was described in the methods? The methods did not include any details about how you decided whether a candidate gene was "near" a QTL.

The assessment of "nearness" was made based on the heuristic QTL intervals, as is now explained in the methods: *Associations between QTLs and candidate genes were proposed when a candidate gene fell within an heuristic QTL interval, as defined earlier.* {201-203}. An analysis of the probability with which these overlaps may have occurred by random chance is included (see **R3A.1**)

R3C.46: Line 244: I don't like the use of the word "historical" to describe these QTLs. I think "previously identified" or "known" QTLs would be more apt.

We understand this objection, but the term "historical QTL" is used widely in the community that is interested in comparative QTL analysis, and it conveys a meaning that is recognizable and clear. The term "known" would imply too much certainty, while "previously identified" is awkward and might be confused with "previously within this study".

Discussion:

R3C.47: Lines 297-30: This sentence is a bit clunky. Consider: "We conducted QTL analyses in five RIL populations, representing nine diverse oat varieties. We identified QTLs for important agricultural

traits...” In addition, I don’t like use of “their potential causes”. I think it would be more accurate to say, “the causal loci potentially underlying trait variation”, or something along those lines.

We revised this to the active voice, and removed chunkiness by breaking the sentence in two: “Here we report a reference-based marker and quantitative trait analyses from five RIL populations representing nine diverse oat varieties. These analyses provide...” {477-480}. We retained the term “potential causes” because it means the same thing as “the causal loci potentially underlying trait variation” and it is a more compact way of expressing the result, such that we can make a short and succinct opening statement to the discussion.

R3C.48: Line 300: Remove “solid”.

Removed {480}.

R3C.49: Lines 308-312: Change to: “Hence, we speculated about...” and “However, due to the small population sizes and lack of a reference genome, we could not conclusively identify the locations or causes of these differences.”

We have revised this sentence similar to what was suggested: “However, due to the small population sizes and the lack of a reference genome, we could not conclusively identify the locations or causes of these differences” {492-494}

R3C.50: Line 316: There is a typo, it should read “unbalanced”.

The sentence containing this word was removed {see **next point**}

R3C.51: Lines 319-20: This is surprising!

The lack of diagrams in the literature showing meiotic pairing in unbalanced translocations was surprising to us too, and remains so. However, Reviewer 1 also questioned this statement (**R1.6**) and we note that the journal instructions suggest not to make statements about novelty. Therefore we have removed these two sentences. The novelty can stand on its own. {formerly located at 501-504}

R3C.52: Line 336: Population structure and local adaptation are not synonymous.

We have clarified through rewording that the first reference examined population structure and the second local adaptation, which we now refer to as breeding history: “...*This appears to have been the case in the oat CORE population*³⁵, where we noted that *chr1A, chr1C, and chr7D contain loci that are highly correlated with population structure*³⁵ that span most or all of these respective chromosomes. *Chr7D (Mrg02) and chr1C (Mrg28) also showed signatures of breeding history*² which included a relatively high number of haplotype associations with heading date...” {517-522}.

R3C.53: Lines 337-338: I’m confused about the statement regarding the “signatures of selection” identified on the chromosomes that are also correlated with population structure. Can you clarify why a large number of haplotypes and “contrasting” (?) haplotype diversity provide evidence for selection.

We have changed “signatures of selection” to “signatures of breeding history”, which would have been a better wording in the original reference {521}. Revisions above {R3C.52} to focus on adaptive associations with heading date also address this confusion. Other signatures of breeding history are elaborated in the reference (Bekele et al, 2018) and are not required to support this statement.

R3C.54: Line 347: Change to: “will make it difficult to identify”.

Revised as suggested {531}

R3C.55: Paragraph starting at line 352: I’m having trouble connecting this speculation with the results in your paper. Do you find any evidence of this?

The reference that we provided for this statement was intended to demonstrate this. We have made this clearer by splitting the sentence as follows: “*The phenomenon of pseudo-linkage may have coupled genes that jointly confer adaptation to specific environments. This appears to be the case with flowering time and winter hardiness, as conferred by genes associated with the 1A-1C translocation*” {537-539}.

R3C.56: Line 389: I do not think that you have done the appropriate statistical analysis to make this statement.

We have now performed an analysis to support this statement (See **R3A.1**).

R3C.57: Lines 393-401: I’m not sure if this discussion (starting at “It is interesting that...”) needs to be included.

We would like to retain this discussion. It is an observation of a phenomenon in oat that will be further emphasized in the Sang companion paper, and it eludes to forthcoming reports of massive chromosome rearrangements in the *Avena* genus. Pointing out this observation here may also be important for readers who are familiar with the Triticeae and expecting to see parallel gene locations in oat. {605-613}

Comments on figures:

R3C.58: Figure 1: There is a typo in the caption, it should read “unbalanced”.

Corrected.

R3C.59: Table 2: Is there any way of representing these pictorially? Perhaps, one could show the LOD plots for each chromosome for all five RILs, at least for a subset of the traits? This may also provide some indication of how well the QTLs replicate across RIL populations. For example, all populations may have a peak in the same region, but some may be underpowered due to smaller sample size.

We have addressed most of this comment where it was raised first in **R3A.2**. The QTL positions are already represented pictorially in Figure 2, and we have made a decision to use this figure primarily for positional interpretation. We have improved the multi-population-based evidence provided by Table 2 with the addition of the “Frequency” column. However, there is no reason to expect all populations to have the same QTL {550-569}, so an extensive treatment of this concept would be misleading. The comparison of full LOD plots for this number of traits and populations would also go well-beyond the already-dense supplemental material, and readers with this level of interest can easily generate their own LOD plots from the data sets that we provide.

Comments on the Supplementary Material:

R3C.60: The Supplementary slides should be reformatted as a supplementary text with supplementary figures that can be referenced.

We have addressed this in **R3B(5)**.

R3C.61: I would recommend using plain text files for all of your supplementary files as these are less easily modified accidentally and more easily accessible. In addition, the supplementary files could be formatted so as to be more accessible. For example, in supplement 7, the QTL mapping results are listed for all populations. However, there is not population column, this information is only buried in the “Attributes” column.

We prefer to keep the Excel-formatted tables (see R3B.5) which is a format recommended by the journal for supplementary data. We are also in the process of submitting the raw marker data to the public T3/Oat database, and the QTL results to the Database GrainGenes. Both databases prefer to receive data in Excel spreadsheets.

The reason that the QTL table lacked a “population” column (and also a “trait” and “environment” column) is that we deliberately chose a standard GFF format for the QTL tables, as explained in {343-345}, which do not allow these as columns. However, to address this comment, we have added three columns and two header rows into the QTL tables in Supplementary Data 5, with instructions to delete these rows and columns if the data are imported as a GFF file into a track within a J-Browse-based viewer.

REVIEWERS' COMMENTS:

Reviewer #1 (Remarks to the Author):

The revisions strengthen the relevance and clarity of the paper.

I have no additional comments or concerns about the paper.

Reviewer #2 (Remarks to the Author):

This manuscript is acceptable.

Reviewer #3 (Remarks to the Author):

My concerns have been mostly addressed.